



# Earth system data cubes unravel global multivariate dynamics

Miguel D. Mahecha[1,2,3], Fabian Gans[1], Gunnar Brandt[4], Rune Christiansen[5], Sarah E. Cornell[6],
Normann Fomferra[4], Guido Kraemer[1,7], Jonas Peters[5], Paul Bodesheim[1,8], Gustau Camps-Valls[7],
Jonathan F. Donges[6,9], Wouter Dorigo[10], Lina Estupiñan-Suarez[1], Victor H. Gutierrez-Velez[11],
Martin Gutwin[1,12], Martin Jung[1], Maria C. Londoño[13], Diego G. Miralles[14], Phillip Papastefanou[15], and
Markus Reichstein[1,2,3]

[1]Max Planck Institute for Biogeochemistry, 07745 Jena, Germany
[2]German Centre for Integrative Biodiversity Research (iDiv), Deutscher Platz 5e, 04103 Leipzig, Germany
[3]Michael Stifel Center Jena for Data-Driven and Simulation Science, 07743 Jena, Germany
[4]Brockmann Consult GmbH, 21029 Hamburg, Germany
[5]Department of Mathematical Sciences, University of Copenhagen, Denmark
[6]Stockholm Resilience Center, Stockholm University, SE-106 91 Stockholm, Sweden
[7]Image Processing Lab, Universitat de València, 46980 Paterna (València), Spain
[8]Computer Vision Group, Computer Science, Friedrich Schiller University, Jena, Germany
[9]Earth System Analysis, Potsdam Institute for Climate Impact Research, PIK, Germany
[10]Department of Geodesy and Geo-Information, TU Wien, Austria
[11]Department of Geography and Urban Studies, Temple University, Philadelphia, USA
[12]Department of Geography, Friedrich Schiller University, Jena, Germany
[13]Alexander von Humboldt Biological Resources Research Institute, Bogotá, Colombia
[14]Laboratory of Hydrology and Water Management, Ghent Unversity, Ghent 9000, Belgium
[15]TUM School of Life Sciences Weihenstephan, Technical University of Munich, 85356 Freising, Germany

**Correspondence:** M.D. Mahecha & F. Gans ({mmahecha,fgans}@bgc-jena.mpg.de)

**Abstract.** Understanding Earth system dynamics in the light of ongoing human intervention and dependency remains a major
scientific challenge. The unprecedented availability of data streams describing different facets of the Earth now offers fun-
damentally new avenues to address this quest. However, several practical hurdles, especially the lack of data interoperability,
limit the joint potential of these data streams. Today many initiatives within and beyond the Earth system sciences are exploring

5    new approaches to overcome these hurdles and meet the growing inter-disciplinary need for data-intensive research; using data
cubes is one promising avenue. Here, we introduce the concept of *Earth system data cubes* and how to operate on them in a for-
mal way. The idea is that treating multiple data dimensions, such as spatial, temporal, variable, frequency and other grids alike,
allows effective application of user-defined functions to co-interpret Earth observations and/or model-data. An implementation
of this concept combines analysis-ready data cubes with a suitable analytic interface. In three case studies we demonstrate how

10   the concept and its implementation facilitate the execution of complex workflows for research across multiple variables, spatial
and temporal scales: 1) summary statistics for ecosystem and climate dynamics; 2) intrinsic dimensionality analysis on mul-
tiple time-scales; and 3) data-model integration. We discuss the emerging perspectives for investigating global interacting and
coupled phenomena in observed or simulated data. Latest developments in machine learning, causal inference, and model data
integration can be seamlessly implemented in the proposed framework, supporting rapid progress in data-intensive research

15   across disciplinary boundaries.



# 1 Introduction

Predicting the Earth system's future trajectory given ongoing human intervention into the climate system and land surface transformations requires a deep understanding of its functioning (Schellnhuber, 1999; Stocker et al., 2013). In particular, it requires unravelling the complex interactions between the Earth's subsystems, often termed as "spheres": atmosphere, biosphere, hydrosphere (including oceans and cryosphere), pedosphere or lithosphere, and increasingly the "anthroposphere". The grand opportunity today is that the many key processes in various subsystems of the Earth are constantly monitored. Networks of ecological, hydro-meteorological and atmospheric in-situ measurements, for instance, provide continuous insights into the dynamics of the terrestrial water and carbon fluxes (Dorigo et al., 2011; Baldocchi, 2014; Wingate et al., 2015; Mahecha et al., 2017). Earth observations retrieved from satellite remote sensing enable a synoptic view of the planet and describe a wide range of phenomena in space and time (Pfeifer et al., 2012; Skidmore et al., 2015; Mathieu et al., 2017). The subsequent integration of in-situ and space-derived data, e.g. via machine learning methods, leads to a range of unprecedented quasi-observational data streams (e.g. Tramontana et al., 2016; Balsamo et al., 2018; Bodesheim et al., 2018; Jung et al., 2019). Likewise, diagnostic models that encode basic process knowledge but which are essentially driven by observations, produce highly relevant data products (see e.g. Duveiller and Cescatti, 2016; Jiang and Ryu, 2016a; Martens et al., 2017; Ryu et al., 2018). Many of these derived data streams are essential for monitoring the climate system including land surface dynamics (see for instance the Essential Climate Variables, ECVs; Hollmann et al., 2013; Bojinski et al., 2014), oceans at different depths (Essential Ocean Variables, EOVs; Miloslavich et al., 2018) or the various aspects of biodiversity (Essential Biodiversity Variables, EBVs; Pereira et al., 2013). Together, these essential variables describe the state of the planet and are indispensable for evaluating Earth system models (Eyring et al., 2019).

In terms of data availability we are well prepared. But can we really exploit this multitude of data streams efficiently and diagnose the state of the Earth system? In principle our answer would be affirmative, but in practical terms we perceive high barriers to efficiently interconnecting multiple data streams and further linking these to data analytic frameworks (as discussed for the EBVs by Hardisty et al., 2019). Examples of these issues are (i) insufficient data discoverability, (ii) access barriers, e.g. restrictive data use policies, (iii) lack of capacity building for interpretation, e.g., understanding the assumptions and suitable areas of application, (iv) quality and uncertainty information, (v) persistency of data sets and evolution of maintained data sets, (vi) reproducibility for independent researchers, (vii) inconsistencies in naming or unit conventions, and (viii) co-interpretability, e.g., either due to spatiotemporal alignment issues, or physical inconsistencies, among others. Some of these issues are relevant to specific data streams and scientific communities only. In most cases, however, these issues reflect the neglect of the FAIR principles (to be *Findable, Accessible, Interoperable, and Re-usable*; Wilkinson et al., 2016). If the lack of FAIR principles and limited (co-)interpretability come together, they constitute a major obstacle in science and slow down the path to new discoveries. Or, to put it as a challenge, we need new solutions that minimize the obstacles that hinder scientists from capitalizing on the existing data streams and accelerate scientific progress. More specifically, we need interfaces that allow for interacting with a wide range of data streams and enable their joint analysis either locally or in the cloud.



As long as we do not overcome data interoperability limitations, Earth system sciences cannot fully exploit the promises of novel data-driven exploration and modelling approaches to answer key questions related to rapid changes in the Earth system (Karpatne et al., 2018; Bergen et al., 2019; Camps-Valls et al., 2019; Reichstein et al., 2019). A variety of approaches has been developed to interpret Earth observations and big-data in the Earth system sciences in general (for an overview see e.g. Sudmanns et al., 2019), and gridded spatiotemporal data as a special case (Nativi et al., 2017; Lu et al., 2018). For the latter, data cubes have become recently popular addressing an increasing demand for efficient access, analysis, and processing capabilities for high-resolution remote sensing products. The existing data cube initiatives and concepts (e.g. Baumann et al., 2016; Lewis et al., 2017; Nativi et al., 2017; Appel and Pebesma, 2019) vary in their motivations and functionalities. Most of the data cube initiatives are, however, motivated by the need for accessing singular (very) high resolution data cubes, e.g. from satellite remote sensing or climate reanalysis, and not by the need for global multivariate data exploitation.

This paper has two objectives: first, we aim to formalize the idea of an *Earth system data cube* that is tailored to explore a variety of Earth system data streams together and thus largely complements the existing approaches. The proposed mathematical formalism intends to illustrate how one can efficiently operate such data cubes. Second, the paper aims at introducing the *Earth System Data Lab* (ESDL, https://earthsystemdatalab.net). The ESDL is an integrated data and analytical hub that curates a multitude of data streams representing key processes of the different subsystems of the Earth in a common data model and coordinate reference system. This infrastructure enables researchers to apply their *user defined functions* (UDFs) to these *analysis-ready data* (ARD). Together, these elements minimize the hurdle to co-explore a multitude of Earth system data streams. Most known initiatives intend to preserve the resolutions of the underlying data and facilitate their direct exploitation, like the Earth Server (Baumann et al., 2016) or the Google Earth Engine (Gorelick et al., 2017). The ESDL, instead, is built around singular data cubes on common spatiotemporal grids, that include a high number of variables as a dimension in its own right. This design principle is thought to be advantageous compared to building data cubes from individual data streams without considering their interactions from the very beginning. We believe that the ESDL, due to its multivariate structure and the easy-to-use interface, is well-suited for being part of data-driven challenges, as regularly organized by the machine learning community, for example.

The reminder of the paper is organized as follows: Sect. 2 introduces the concept based on a formal definition of *Earth System Data Cubes* and explains how user defined functions can interact with them. In Sect. 3, we describe the implementation of the *Earth System Data Lab* in the programming language Julia and as a cloud based data hub. Sect. 4 then illustrates three research use cases that highlight different ways to make use of the ESDL. We present an example from an univariate analysis, characterizing seasonal dynamics of some selected variables; an example from high-dimensional data analysis; and an example for the representation of a model-data-integration approach. In Sect. 5, we discuss the current advantages and limitations of our approach and put an emphasis on required future developments.



## 2 Concept

Our vision is that multiple spatiotemporal data streams shall be treated as a singular, yet potentially very high-dimensional data stream. We call this singular data stream an *Earth System Data Cube*. For the sake of clarity, we introduce a mathematical representation of the Earth system data cube and define operations on it. Further details on an efficient implementation are provided in Sections 3.2 and 3.3.

Suppose we observe $p$ variables $Y^1, \ldots, Y^p$, each under a (possibly different) range of conditions. A first step towards data integration is to (re)sample all data streams onto a common domain $J$ (e.g., a spatiotemporal grid) to obtain the indexed set $\{(Y_j^1, \ldots, Y_j^p)\}_{j \in J}$ of multivariate observations. However, when calculating simple variable summaries, or performing spatio-temporal aggregations of the data, it is often computationally obstructive to discriminate between observations obtained from different variables. We therefore propose to consider the "variable indicator" $k \in \{1, \ldots, p\}$ as simply another dimension of the index set, and view the data as the collection $\{X_i\}_{i \in I}$ of *univariate* observations, where $I = J \times \{1, \ldots, p\}$ and where $X_{(j,k)} := Y_k^j$. With this idea in mind, we now formally define the *Earth System Data Cube* (short *data cube*).

A data cube $C$ consists of a triplet $(L, G, X)$ of components to be described below.

- $L$ is a set of labels, called dimensions, describing the axes of the data cube. For example, $L = \{lat, lon, time, var\}$ describes a data cube containing spatiotemporal observations from a range of different variables. The number of dimensions $|L|$ is referred to as the *order* of the cube $C$, in the above example, $|L| = 4$.

- $G$ is a collection $\{\mathrm{grid}(\ell)\}_{\ell \in L}$ of grids along the axes in $L$. For every $\ell \in L$, the set $\mathrm{grid}(\ell)$ is a discrete subset of the domain of the axis $\ell$, specifying the resolution at which data is available along this axis. Every set $\mathrm{grid}(\ell)$ is required to contain at least two elements. Dimensions containing only one grid point are dropped. The collection $G$ defines the hyperrectangular index set $I(G) := \bigtimes_{\ell \in L} \mathrm{grid}(\ell)$, motivating the name "cube". For example,

$$
\begin{aligned}
I(G) &= \bigtimes_{\ell \in L} \mathrm{grid}(\ell) \\
&= \mathrm{grid}(lat) \times \mathrm{grid}(lon) \times \mathrm{grid}(time) \times \mathrm{grid}(var) \\
&= \{-89.75, \ldots, 89.75\} \times \{-179.75, \ldots, 179.75\} \times \{01.01.2010, \ldots, 31.12.2010\} \times \{\mathrm{GPP}, \mathrm{SWC}, \mathrm{R}_g\} \\
&= \{(-89.75, -179.75, 01.01.2010, \mathrm{GPP}), \ldots, (89.75, 179.75, 31.12.2010, \mathrm{R}_g)\}.
\end{aligned}
$$

Since $G$ and $I(G)$ are in one-to-one correspondence, we will use the two interchangeably.

- $X$ is a collection of data $\{X_i\}_{i \in I(G)} \subseteq \mathbb{R}_{\mathrm{NA}} := \mathbb{R} \cup \{\mathrm{NA}\}$ observed at the grid points in $I(G)$.

In this view, the data can be treated as a collection $\{X_i\}_{i \in I(G)}$ of *univariate* observations, even if they encode different variables. In the above example the variable axis is a nominal grid with the entries GPP (gross primary production), SWC (soil water content), and $\mathrm{R}_g$ (global radiation). The set of all data cubes with dimensions $L$ will be denoted by $\mathcal{C}(L)$. Data cubes that contain one variable only are can be considered as special case; other common choices of $L$ are described in Table 1. The list of example axes labels used in the table is, of course, not exhaustive. Other relevant dimensions could be, for example, model versions, model parameters, quality flags, or uncertainty estimates. Note that *by definition*, a data cube only depends on



its dimensions through the *set* of axes $L$, and is therefore indifferent to any order of these. In the remainder of this article, the notion of data cubes refers to this concept.

**Table 1.** Typical sets of data cubes $\mathcal{C}(L)$ of varying orders $|L|$ with characteristic dimensions $L$.

| Order $|L|$ | Set of data cubes $\mathcal{C}(L)$ | Description of $\mathcal{C}(L)$ |
|---|---|---|
| 0 | $\mathcal{C}(\{\})$ | Scalar value where no dimension is defined. |
| 1 | $\mathcal{C}(\{lat\})$ | Univariate latitudinal profile. |
| 1 | $\mathcal{C}(\{lon\})$ | Univariate longitudinal profile. |
| 1 | $\mathcal{C}(\{time\})$ | Univariate time series. |
| 1 | $\mathcal{C}(\{var\})$ | Single multivariate observation. |
| 2 | $\mathcal{C}(\{lat, lon\})$ | Univariate static geographical map. |
| 2 | $\mathcal{C}(\{lat, time\})$ | Univariate Hovmöller diagram: zonal pattern over time. |
| 2 | $\mathcal{C}(\{lat, var\})$ | Multivariate latitudinal profile. |
| 2 | $\mathcal{C}(\{lon, time\})$ | Univariate Hovmöller diagram: meridional pattern over time. |
| 2 | $\mathcal{C}(\{lon, var\})$ | Multivariate longitudinal profile. |
| 2 | $\mathcal{C}(\{time, var\})$ | Multivariate time series. |
| 2 | $\mathcal{C}(\{time, freq\})$ | Univariate time frequency plane. |
| 3 | $\mathcal{C}(\{lat, lon, time\})$ | Univariate data cube. |
| 3 | $\mathcal{C}(\{lat, lon, var\})$ | Multivariate map, e.g. a global map of different soil properties. |
| 3 | $\mathcal{C}(\{lat, time, var\})$ | Multivariate latitudinal Hovmöller diagram. |
| 3 | $\mathcal{C}(\{lon, time, var\})$ | Multivariate longitudinal Hovmöller diagram. |
| 3 | $\mathcal{C}(\{time, freq, var\})$ | Multivariate spectrally decomposed time series. |
| 4 | $\mathcal{C}(\{lat, lon, time, var\})$ | Multivariate spatiotemporal cube. |
| 4 | $\mathcal{C}(\{lat, lon, time, freq\})$ | Univariate spectrally decomposed data cube. |
| 5 | $\mathcal{C}(\{lat, lon, time, var, ens\})$ | Multivariate ensemble of model simulations. |



## 2.1 Operations on an Earth System Data Cube

To exploit an Earth system data cube efficiently, scientific workflows need to be translated into operations executable on data cubes as described above. More specifically, the output of each operation on a data cube should yield another data cube. The entire workflow of a project, possibly a succession of analyses performed by different collaborators, can then be expressed as a composition of several *user defined functions* (UDFs) performed on a single (input-) data cube. Besides unifying all statistical data analyses into a common concept, the idea of expressing workflows as functional operations on data cubes comes with

another important advantage: as soon as a workflow is implemented as a suitable set of UDFs, it can be reused on any other sufficiently similar data cube to produce the same kind of output.

In its most general form, a user defined function $C \mapsto f(C)$ operates by (i) extracting relevant information from $C$, (ii) performing calculations on the extracted information, and (iii) storing these calculations into a new data cube $f(C)$. In order to perform step (i), $f$ expects a minimal set of dimensions $E$ of the input cube. The returned set of axes for an input cube with

dimensions $E$ will be denoted by $R$. That is, $f$ is a mapping such that

$$f : \mathcal{C}(E) \to \mathcal{C}(R). \tag{1}$$

Alongside the function $f$, one has to define the sets $E$ and $R$ which we will refer to as *minimal input-* and *minimal output dimensions*, respectively.

A major advantage of thinking in data cube workflows is that low-dimensional functions can be applied to higher-dimensional

cubes by simple functional extensions: a function can be acting along a particular set of dimensions while looping across all unspecified dimensions. For example, the function that computes the temporal mean of a univariate time series should allow for an input data cube, which, in addition to a temporal grid, contains spatial information. The output of such an operation should then be a cube of spatially gridded temporal means. Similarly, the function should be applicable to cubes containing multivariate observations. Here, we expect the output to contain one temporal mean per supplied variable. In general, a function

$f$ defined on $\mathcal{C}(E)$ should naturally extend to any set $\mathcal{C}(E \cup A)$ with $A \cap R = \emptyset$ by executing the described "apply"-operation. The code package accompanying this paper (described in Sect. 3) automatically equips every UDF with such a functionality. A schematic description of this approach is illustrated in Fig. 1.

The approach outlined above is very convenient to describe workflows, i.e. recursive chains of UDFs. Let $f_1, \ldots f_n$ be a sequence of UDFs with corresponding minimal input/output dimensions $(E_1, R_1), \ldots, (E_n, R_n)$. If an output dimension $R_i$ is

a subset of subsequent input $E_{i+1}$ we can chain these functions. A *recursive workflow* emerges when $R_i \subseteq E_{i+1}$ for all $i$, by iteratively chaining $f_1, \ldots, f_n$ upon one another. The input/output dimensions of the resulting cube are $(E_1, R_n)$.

Overall, the definition of an Earth system data cube and associated operations on it do not only guide the implementation strategy, but also help us summarize potentially complicated analytic procedures in a common language. For the sake of readability, in the following, we will not distinguish between a function $f$ (defined only for minimal input) and its extension $\bar{f}$

(equipped with the apply-functionality, see Fig. 1). The former will be referred to as an *atomic* function. We typically indicate the minimal input/output dimensions $(E, R)$ of a function $f$ by writing $f_E^R$. Since the pair $(E, R)$ does not determine the mapping $f$, this notation should not be understood as the parameterization of a function class, but rather provide an easy way





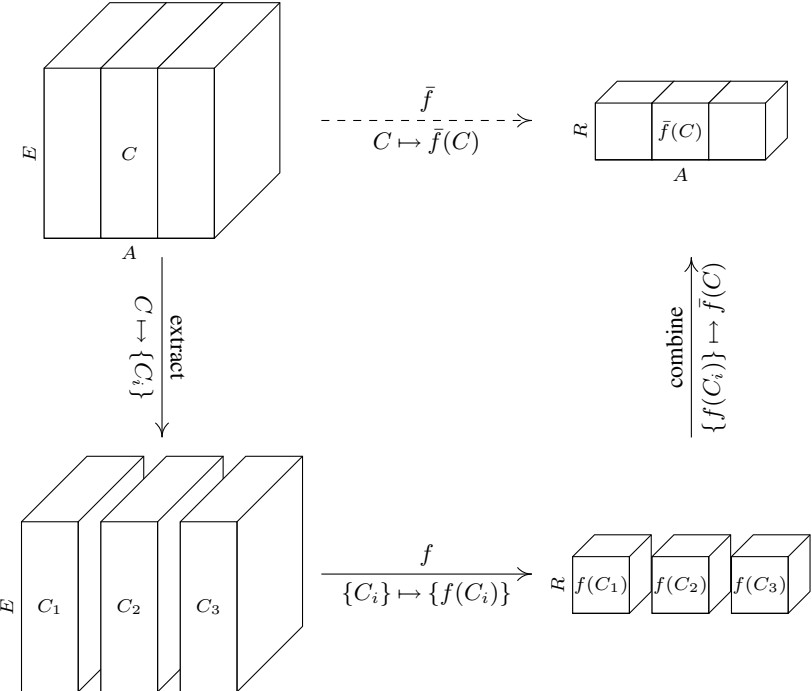

**Figure 1.** Schematic illustration of the "apply"-functionality: A function $f : \mathcal{C}(E) \to \mathcal{C}(R)$ is extended to the set of cubes with dimensions $E \cup A$, where $A$ is an arbitrary set of dimensions with $A \cap R = \emptyset$. Given a cube $C \in \mathcal{C}(E \cup A)$, the extension $\bar{f}(C)$ is constructed by iterating over all grid points $i$ along the dimensions in $A$ to obtain the collection $\{C_i\} \subseteq \mathcal{C}(E)$ of sliced cubes, applying $f$ to every cube $C_i$ separately, and binding the collection $\{f(C_i)\}$ into the output cube $\bar{f}(C) \in \mathcal{C}(R \cup A)$. Here, the index $i$ runs through all elements in $\bigtimes_{a \in A} \text{grid}(a)$.

to perform input-control, and to anticipate the output dimensions of a cube returned by $f$. For instance, following the discussion above, a function denoted by $f_E^R$ can be applied to any cube with dimension $E \cup A$ satisfying that $A \cap R = \emptyset$, and returns a

cube with dimensions $R \cup A$. To avoid ambiguities, additional notation is needed when distinguishing between two functions with the same pair of minimal input/output dimensions.

## 2.2 Examples

In the following, we present some special operations that are routinely needed in explorations of Earth system data cubes:

*Reducing* describes a function that calculates some scalar measure (e.g. the sample mean). Consider, for instance, the need

to estimate the mean of a univariate data cube, of course weighted by the area of the spatial grid cells. An operation of this kind expects a cube with dimensions $E = \{lat, lon, time\}$ and returns a cube with dimensions $R = \{\}$, and is therefore a mapping

$$f_{\{lat,lon,time\}}^{\{\}} : \mathcal{C}(\{lat, lon, time\}) \to \mathcal{C}(\{\}). \tag{2}$$





This mapping can now be applied to any data cube of potentially higher (but not lower) dimensionality. For instance, $f$ is automatically extended to a multivariate spatio-temporal data cube (Table 1) with the mapping

$$f^{\{\}}_{\{lat,lon,time\}} : \mathcal{C}(\{lat,lon,time,var\}) \rightarrow \mathcal{C}(\{var\}), \tag{3}$$


which computes one spatio-temporal mean for each variable.

*Cropping* is sub-setting a data cube while maintaining the order of a cube. A cropping operation typically reduces certain axes of a data cube to only contain specified grid points (and therefore requires the input cube to contain these grid points). For instance, a function that extracts a certain "cropped" fraction $T_0$ along the temporal cover, expects an input cube containing a
*time*-axis with a grid at least as highly resolved as $T_0$. This function preserves the dimensionality of the cube, but reduces the grid along the time axis, i.e.,

$$f^{\{time\}}_{\{time\}} : \mathcal{C}(\{time\} \,|\, \mathrm{grid}(time) \supseteq T_0) \rightarrow \mathcal{C}(\{time\} \,|\, \mathrm{grid}(time) = T_0), \tag{4}$$

where we have used $\mathcal{C}(L\,|\,P)$ to denote the set of cubes with dimensions $L$ satisfying the condition $P$. Thanks to the apply functionality, this atomic function can be used on any cube of higher order. For example, it is readily extended to a mapping


$$f^{\{time\}}_{\{time\}} : \mathcal{C}(\{lat,lon,time\} \,|\, \mathrm{grid}(time) \supseteq T_0) \rightarrow \mathcal{C}(\{lat,lon,time\} \,|\, \mathrm{grid}(time) = T_0), \tag{5}$$

which crops the time axis of cubes with dimensions $\{lat,lon,time\}$. Analogously, all dimensions can be subsetted as long as the length of the dimension is larger than one. The latter would be called slicing.

*Slicing* refers to a subsetting operation in which a dimension of the cube is degenerated, and the order of the cube is reduced and can be interpreted as a special from of cropping. For instance, if we only select a singular time-instance $t_0$, the time
dimension effectively vanishes as we do not longer need a vector spaced dimension to represent its values. When applied to a spatio-temporal data cube, this amounts to a mapping

$$f^{\{\}}_{\{time\}} : \mathcal{C}(\{lat,lon,time\} \,|\, \mathrm{grid}(time) \ni t_0) \rightarrow \mathcal{C}(\{lat,lon\}). \tag{6}$$

*Expansions* are operations where the order of the output cube is higher than the order of the corresponding input cube. A discrete spectral decomposition of time series, for example, generates a new dimension with characteristic frequency classes:


$$f^{\{time,freq\}}_{\{time\}} : \mathcal{C}(\{time\}) \rightarrow \mathcal{C}(\{time,freq\}). \tag{7}$$

*Multiple Cube Handling* is often needed, for instance when fitting a regression model where response and predictions are stored in different cubes. Also, we may be interested in outputting the fitted values and the residuals in two separate cubes. This amounts to an atomic operation

$$f^{\{para\},\{time\}}_{\{time,var\},\{time\}} : \mathcal{C}(\{time,var\}) \times \mathcal{C}(\{time\}) \rightarrow \mathcal{C}(\{para\}) \times \mathcal{C}(\{time\}), \tag{8}$$

which expects a multivariate data cube for the predictors $C_1 \in \mathcal{C}(\{time,var\})$, and a univariate cube for the targets $C_2 \in \mathcal{C}(\{time\})$. The output consists of a vector of fitted parameters $\tilde{C}_1 \in \mathcal{C}(\{para\})$ and a residual time series $\tilde{C}_2 \in \mathcal{C}(\{time\})$ to compute the model performance. This concept also allows the integration of more than two input and/or output cubes.





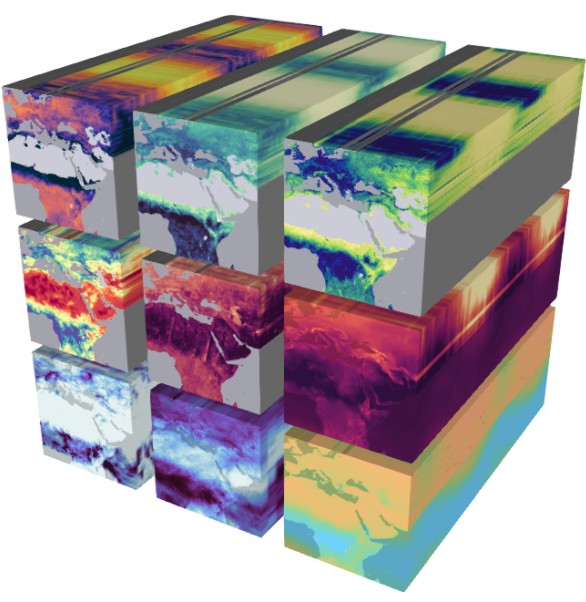

**Figure 2.** Visualization of the implemented Earth system data cube (an animation is provided online https://youtu.be/9L4-fq48Ev0). The figure shows from the top left to bottom right the variables sensible heat (H), latent heat (LE), gross primary production (GPP), surface moisture (SM), land surface temperature (LST), air temperature (Tair), cloudiness (C), precipitation (P), and water vapour (V). References to the individual data sources are given in Appendix 1. Here, the resolution in space is 0.25°, in time 8-days, and we are inspecting the time from May 2008 to May 2010, the spatial range is from 15°S to 60°N, and 10°E to 65°W.

## 3 Data streams and implementation

The concept as described in Sect. 2 is generic, i.e. independent of the implemented Earth system data cube and of the technical
solution of the implementation. In the following, we firstly present the data used in our implementation of the ESDL which is available online, and secondly describe the implementation strategy we developed in this project.

### 3.1 Data streams in the ESDL

The data streams included so far were chosen to enable research on the following topics (a complete list is provided in Appendix A):

(i) *Ecosystem states* at the global scale in terms of relevant biophysical variables. Examples are, for instance, leaf area index (LAI), the fraction of photosynthetically active radiation (FAPAR), and albedo (Disney et al., 2016; Pinty et al., 2006; Blessing and Löw, 2017).





(ii) *Biosphere-atmosphere interactions* as encoded in land fluxes of $CO_2$ i.e. GPP, terrestrial ecosystem respiration ($R_{eco}$), and the net ecosystem exchange (NEE) as well as the latent heat (LE) and sensible heat (H) energy fluxes. Here, we rely
mostly on the FLUXCOM data suite (Tramontana et al., 2016; Jung et al., 2019).

(iii) *Terrestrial hydrology* requires a wide range of variables. We mainly ingest data from the Global Land Evaporation Amsterdam Model (GLEAM; Martens et al., 2017; Miralles et al., 2011) which provides a series of relevant surface hydrological properties such as surface (SM) and root-zone soil moisture (SMroot), but also potential evaporation (Ep) and evaporative stress (S) conditions, among others. Ingesting entire products such as GLEAM ensures internal consistency.

(iv) *State of the atmosphere* is described using data generated by the Climate Change Initiative by the ESA (CCI) in terms of aerosol optical thickness at different wavelength (AOD550, AOD555, AOD659, and AOD1610; Holzer-Popp et al., 2013), total ozone column (Van Roozendael et al., 2012; Lerot et al., 2014), as well as surface ozone (which is more relevant to plants), and total column water vapour (TCWV; Schröder et al., 2012; Schneider et al., 2013).

(v) *Meteorological conditions* are described via the reanalysis data i.e. the ERA5 product. Additionally, precipitation is
ingested from the Global Precipitation Climatology Project (GPCP; Adler et al., 2003; Huffman et al., 2009).

Together, these data streams form data cubes of intermediate spatial and temporal resolutions (0.25°, 0.083°; both 8-daily). These variables described here are described in more detail in a list provided in Appendix A, which may, however, already be incomplete at the time of publication, as the ESDL is a living data suite, constantly expanding according to users' requests. For the latest overview, we refer the reader to the website (https://www.earthsystemdatalab.net/).

Moreover, to show the portability of the approach, we have developed a regional data cube for Colombia. This work supports the Colombian Biodiversity Observational Network activities within GEOBON. This regional data cube has a 1km (0.083°) resolution and focuses on remote sensing derived data products (i.e. LAI, FAPAR, the normalized difference vegetation index NDVI, the enhanced vegetation index EVI, LST, and burnt area). In addition to the global ESDL, monthly mean products such as cloud cover (Wilson and Jetz, 2016) have been ingested given of their recurrent applicability in biodiversity studies
at regional scales. Data layers from governmental organizations providing detailed information about ecosystems are also available that allow a national characterization and deeper understanding of ecosystem changes by natural or human drivers. These are the national ecosystem map (IDEAM et al., 2017), biotic units map (Londoño et al., 2017), wetlands (Flórez et al., 2016) and agriculture frontier maps (MADR-UPRA, 2017). Additionally, GPP, evapotranspriation, shortwave radiation, PAR and diffuse PAR from the Breathing Earth System Simulator (BESS; Ryu et al., 2011; Jiang and Ryu, 2016b; Ryu et al., 2018)
and albedo from QA4ECV (http://www.qa4ecv.eu/) are available, among others. This regional Earth system data cube should serve as a platform for analysis in a region with variability of landscape, high biodiversity, ecosystem transitions gradients and facing rapid land use change (Sierra et al., 2017).



### 3.2 Implementation

To put the concept of an Earth system data cube as outlined in Sect. 2 into practice, a co-author of this paper (FG) developed
an implementation in the relatively young scientific programming computing language Julia (julialang.org; Bezanson et al.,
2017) and developed the `ESDL.jl` package. The goal was that the user does not have to explicitly deal with the complexities
of sequential data input/output handling and can concentrate on implementing the atomic functions and workflows, while the
system takes care of necessary out of core and out of memory computations. The following is a sketched description of the prin-
ciples of the Julia-based `ESDL.jl` implementation. We choose Julia to translate the concepts outlined into efficient computer
code because it has clear advantages for data cube applications besides its general elegance in scientific computing in terms of
speed, dynamic programming, multiple dispatch, and syntax (Perkel, 2019). Specifically, Julia allows for generic processing
of high-dimensional data without large code repetitions. At the core of the Julia `ESDL.jl` toolbox are the `mapslices` and
`mapCube` functions, which execute user-defined functions on the data cube as follows:

- Given some large data cube $C = (L, G, X)$, the `ESDL` function `subsetcube(C)` will retrieve a handle to $C$ that fully
  describes $L$ and $G$.

- Knowledge on the desired $L$ and $G$ allows us to develop a suitable user defined function $f_E^R$.

- Depending on the exact needs, `mapslices` and `mapCube` will then be used to apply the $f_E^R$ on a cube as illustrated
  in Fig. 1. `mapCube` is a strict implementation of the cube mapping concept described here, where it is mandatory to
  explicitly describe $E$ and $R$ such that the atomic function is fully operational. `mapslices` is a convenient wrapper
  around the `mapCube` function that tries to impute the output dimensions given the user function definition to ease the
  application of the functions where the output dimensions are trivial. Internally, `mapslices` and `mapCube` verify that
  $E \subseteq L$ and other conditions.

The case studies developed in Sect. 4 are accompanied with code that illustrates this approach in practice.

Of course there are also alternatives to Julia. Lu et al. (2018) recently reviewed different ways of applying functions on array
data sets in R, Octave, Python, Rasdaman and SciDB. One requirement of such a mapping function is that it should be scalable,
which means that it should process data larger than the computer memory and, if needed, in parallel. While existing solutions
are sufficient for certain applications, most are not consistent with the cube mapping concept as described in Sect. 2. For
instance, the required handling of complex workflows of multiple cubes (Eq. 8) is typically not possible in the existing solutions
that have been reviewed. In some cases, issues in the computational efficiency of the underlying programming languages render
certain solutions not suitable. This is particular the case when user-defined functions become complex. Likewise, certain
properties such as the desired indifference to the ordering in axes dimensions are often not foreseen. One suitable alternative
to Julia is available in Python. The `xarray` (http://xarray.pydata.org) and `dask` packages have been successfully utilised in
the Open Data Cube, Pangeo, and xcube initiatives. An Extensive descriptions on how to work in the ESDL with both Python
and Julia can be accessed from the website: earthsystemdatalab.net.





## 3.3 Storage and processing of the data cube

The ESDL has been built as a generic tool. It is prepared to handle very large volumes of data. Storage techniques for large raster geo-data are generally split into two categories: Database-like solutions like Rasdaman (Baumann et al., 1998) or SciDB (Stonebraker et al., 2013) access data directly through file formats that follow metadata conventions like HDF5 (https://www.hdfgroup.org/) or NetCDF (https://www.unidata.ucar.edu/software/netcdf/). Database solutions shine in settings where multiple users repeatedly request (typically small) subsets of data cube, which might not be rectangular, because the database can accelerate access by adjusting to common access patterns. However, for batch-processing large portions of a data cube, every data entry is ideally accessed only once during the whole computation. Hence, when large fractions of some data cube have to be accessed, users will usually avoid the overhead of building and maintaining a database and rather aim for accessing the data directly from its files. This experience is often perceived as more "natural" for Earth system scientists who are used to "touching" their data, knowing where files are located, and so forth. Databases instead offer, by construction, an entry point to an otherwise unknown data set.

One disadvantage of the traditional file formats used for storing gridded data is that their data chunks are contained in single files that may become impossible to handle efficiently. This is not problematic when the data is stored on a regular file system where the file format library can read only parts of the file. In cloud-based storage systems it is not common to have an API for accessing only parts of an object, so these file formats are not well suited for being stored in the cloud. Recently, novel solutions for this issue were proposed, including modifications to existing storage formats, e.g. HDF5 cloud, or cloud-optimized GeoTiff, among others, as well as completely new storage formats, in particular Zarr (https://zarr.readthedocs.io/) and TileDB (https://tiledb.io/). While working with these formats is very similar to traditional solutions (like HDF5 and NetCDF), these new formats are optimized for cloud storage as well as for parallel read and write operations. Here we chose to use the new Zarr format. The reason is that it enables us to share the data cube through an object storage service, where the data is public and can be analyzed directly. Python packages for accessing and analyzing large $N$-dimensional data sets like `xarray` and `dask`, which make a wide range of existing tools readily usable on the cube, and a Julia-approach to read Zarr data have been implemented as well.

At present, the ESDL provides the same data cube in different spatial resolutions and different chunkings of the data speed up data access. In chunked data formats, a large dataset is split into smaller chunks, that can be seen as separate entities where each chunk is represented by an object in an object store. There are several ways to chunk a data cube. Consider the case of a multivariate spatiotemporal cube $\mathcal{C}(\{lat, lon, time, var\})$. One common strategy would to treat every spatial map of each variable and time point as one chunk, which would result in a chunk size of $|grid(lat)| \times |grid(lon)| \times 1 \times 1$. However, because an object can only be accessed as a whole, the time for reading a slice of an univariate data cube does not directly scale with the number of data points accessed, but rather with the number of accessed chunks. Reading out a univariate time series of length 100 from this cube would require accessing 100 chunks. If one would store the same data cube where complete time series are contained in one chunk, one could perform much faster read operations. Table 2 shows an overview of the implemented chunkings for different cubes in the current ESDL environment.





**Table 2.** Resolutions and chunkings of the currently implemented global Earth system data cube per variable. Here, the cubes with chunk size 1 in the time coordinate are optimized for accessing global maps at a time while the other cubes are more suited for processing time-series or regional subsets of the data cube. The cubes are currently hosted on the Object Storage Service by the Open Telecom Cloud under https://obs.eu-de.otc.t-systems.com/obs-esdc-v2.0.0/ (state: Sept. 2019).

| | Chunk size along axis | | |
| --- | --- | --- | --- |
| Resolution | grid($time$) | grid($lat$) | grid($lon$) |
| $0.083°$ | 184 | 270 | 270 |
| $0.083°$ | 1 | 2160 | 4320 |
| $0.25°$ | 184 | 90 | 90 |
| $0.25°$ | 1 | 720 | 1440 |

## 4 Experimental case studies

The overarching motivation for building an Earth system data cube is to support the multifaceted needs of Earth system sciences. Here, we briefly describe three case studies of varying complexity (estimating seasonal means per latitude, dimensionality reduction, and model-data integration) to illustrate how the concept of the Earth system data cube can be put into practice. Clearly, these examples emerge from our own research interest, but the concepts should be portable across different branches of science (the code for producing the results on display are provided as Jupyter notebooks at https:
//github.com/esa-esdl/ESDLPaperCode.jl).

### 4.1 Inspecting summary statistics of biosphere/atmosphere interactions

Data exploration in the Earth system sciences typically starts with inspecting summary statistics. Global mean patterns across variables can give an impression on the long-term behaviour system behaviour in space or time. In this first use case, we aim to describe mean seasonal dynamics of multiple variables across latitudes.

Consider an input data cube of the form $\mathcal{C}(\{lat, lon, time, var\})$. The first step consists in estimating the median seasonal cycles per grid cell. This operation creates a new dimension encoding the "day of year" ($doy$) as described in the atomic function of Eq. (9):

$$f_{\{time\}}^{\{doy\}} : \mathcal{C}(\{lat, lon, time, var\}) \rightarrow \mathcal{C}(\{lat, lon, doy, var\}). \tag{9}$$

In a second step, we apply an averaging function that summarizes the dynamics observed at all longitudes:

$$f_{\{lon\}}^{\{\}} : \mathcal{C}(\{lat, lon, doy, var\}) \rightarrow \mathcal{C}(\{lat, doy, var\}). \tag{10}$$

The result is a a cube of the form $\mathcal{C}(\{lat, doy, var\})$ describing the seasonal pattern of each variable per latitude. Fig. 3 visualizes this analysis for data on gross primary production (GPP), air temperature (Tair), and surface moisture (SM; all





**Figure 3.** Polar diagrams of median seasonal patterns per latitude (land only). The values of the variables are displayed as grey gradient and scale with the distance to the centroid. For each latitude we have a median seasonal cycle specified with the central color code. The left columns shows the patterns for the Northern Hemisphere; the right columns are the analogous figures for the Southern Hemisphere. Here we show the patterns for gross primary production (GPP), air temperature at 2m (Tair), and surface moisture (SM).





references for data streams used are provided in Appendix A). The first row visualizes GPP, on the left side we see the Northern Hemisphere where darker colors describe higher latitudes and the background is the actual value of the variable.

Together, the left and right plots describe the global dynamics of phenology, often referred to as "green wave" (Schwartz, 1998). We clearly see the almost non-existent GPP in high-latitude winters and also find the imprint of constantly low to intermediate productivity values at latitudes that are characterized by dry ecosystems. Pronounced differences between northern and Southern Hemispheres reflect the very different distribution of productive land surface.

For temperature, the observed seasonal dynamics are less complex. We essentially find the constantly high temperature con-
ditions near the equator and visualize the pronounced seasonality at high latitudes. However, fig. 3 also shows that temperature peaks lag behind the June/December solstices in the Northern Hemisphere, while at the Southern Hemispheres the asymmetry of the seasonal cycle in temperature is less pronounced. While the seasonal temperature gradient is a continuum, surface moisture shows a much more complex pattern across latitudes as reflected in summer/winter depressions in certain mid latitudes. For instance, a clear drop at e.g. latitudes of approx. 60°N and even stronger depressions in latitudinal bands dominated by dry
ecosystems.

This example analysis is intended to illustrate how the sequential application of two basic functions on this Earth system data cube can unravel global dynamics across multiple variables. We suspect that applications of this kind can lead to new insights into apparently known phenomena, as they allow to investigate a large number of data streams simultaneously and with consistent methodology.

**4.2  Intrinsic dimensions of ecosystem dynamics**

The main added value of the ESDL approach is its capacity to jointly analyze large numbers of data streams in integrated workflows. A long standing question arising when a system is observed based on multiple variables is whether these are all necessary to represent the underlying dynamics. The question is whether the data observed in $Y \in \mathbb{R}^M$ could be described with a vector space of much smaller dimensionality $Z \in \mathbb{R}^m$ (where $m \ll M$), without loss of information, and what value this
"intrinsic dimensionality" $m$ would have (Lee and Verleysen, 2007; Camastra and Staiano, 2016). Note that in this context the term "dimension" has a very different connotation compared to the "cube dimensions" introduced above.

When thinking about an Earth system data cube, the question about its intrinsic dimensionality could be interrogated along the different axes. In this study we ask if the multitude of data streams, grid($var$), contained in our Earth system data cube is needed to grasp the complexity of the terrestrial surface dynamics. If the compiled data streams were highly redundant, it could
be sufficient to concentrate on only a few orthogonal variables and design the development of the study accordingly. Starting from a cube $\mathcal{C}(\{lat, lon, time, var\})$, we ask at each geographical coordinate if the local vector space spanned by the variables can be compressed such that $m_{var} \ll |\text{grid}(var)|$.

Estimating the intrinsic dimension of high-dimensional datasets has been a matter of research for multiple decades, and we refer the reader to the existing reviews on the subject (e.g. Camastra and Staiano, 2016; Karbauskaite and Dzemyda, 2016).
An intuitive approach is to measure the compressibility of a dataset via dimensionality reduction techniques (see e.g. van der Maaten et al., 2009; Kraemer et al., 2018). In the simplest case, one can apply a principal component analysis (PCA, using



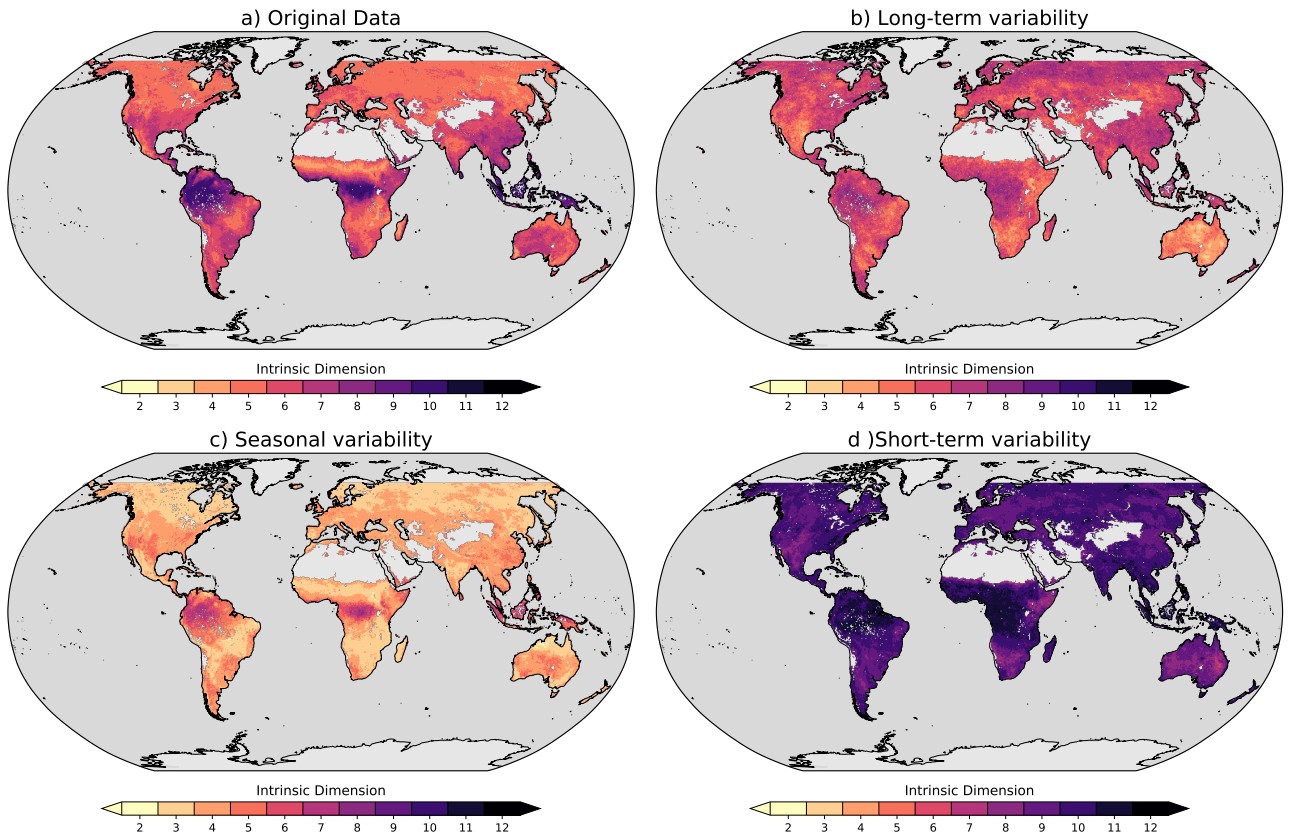

**Figure 4.** Intrinsic dimension of 18 land ecosystem variables. The intrinsic dimension is estimated by counting how many principal components would be needed to explain at least 95% of the variance in the Earth system data cube. The results for the original data is shown in a). The analysis is then repeated based on subsignals of each variable, representing different timescales. In b) we show the intrinsic dimension of long-term modes of variability, in c) for modes representing seasonal components, and d) for modes of short-term variability.

different time points as different observations) and estimate the number of components that together explain a predefined threshold of the data variance. In our application, we followed this approach and chose a threshold value of 95% of variance. The atomic function needed for this study is described in Eq. (11):

$$f_{\{time,var\}}^{\{\}} : \mathcal{C}(\{lat,lon,time,var\}) \to \mathcal{C}(\{lat,lon\}). \tag{11}$$

The output is a map of spatially varying estimates of intrinsic dimensions $m_{var}$. We performed this study considering the following 18 variables relevant to describing land surface dynamics: GPP, $R_{eco}$, NEE, LE, H, LAI, fAPAR, black and white sky albedo (the latter two each from two different sources), SMroot, S, transpiration, bare soil evaporation, evaporation, net radiation, and LST.





Figure 4 shows the results of this analysis for the original data, where the visualized range of intrinsic dimensions ranges from 2 to 13 (the analysis very rarely returns values of 1). At first glance, we find that ecosystems near the equator are of higher intrinsic dimension (up to values of 12) compared to the rest of the land surface. In regions where we expect pronounced seasonal patterns the intrinsic dimensionality is apparently low. We can describe these patterns by 4–7 dimensions. One explanation is that in cases where seasonal cycles control ecosystem dynamics, much of the surface variables tend to co-vary. This

alignment implies that one can represent the dominant source of variance with few components of variability. In regions where the seasonal cycles play only a marginal role other sources of variability dominate that are, however, largely uncorrelated.

In order to verify that seasonality is the main source of variability, we extend the workflow by decomposing each time series (by variable and spatial location) into a series of subsignals via a discrete Fast Fourier Transform (FFT). We then binned the subsignals into short-term, seasonal, and long-term modes of variability (as in Mahecha et al., 2010a), which leads to an

extended data cube as we shown in Eq. (12).

$$f_{\{time\}}^{\{time,freq\}} : \mathcal{C}(\{lat, lon, time, var\}) \to \mathcal{C}(\{lat, lon, time, var, freq\}). \tag{12}$$

The resulting cube is then further processed in Eq. (13) (which is the analogue to Eq. (11)) to extract the intrinsic dimension per time scale:

$$f_{\{time,var\}}^{\{\}} : \mathcal{C}(\{lat, lon, time, var, freq\}) \to \mathcal{C}(\{lat, lon, freq\}). \tag{13}$$

The timescale specific intrinsic dimension estimates only partly confirms the initial conjecture (Fig. 4). Short-term modes of variability always show relatively high intrinsic dimensions, i.e. the high-frequency components in the variables are rather uncorrelated. This finding can either be a hint that we are seeing a set of independent processes, or simply mean noise contamination. Seasonal modes, indeed, are of low intrinsic dimensionality, but considering that these modes are driven essentially by solar forcing only, they are surprisingly high dimensional. Additionally, we find a clear gradient from the inner tropics to

arid and Northernmost ecosystems. Warm and wet ecosystems seem to be characterized by a complex interplay of variables even when analyzing their seasonal components only. One reason could again be that their relevance is rather minor and their temporal evolution is rather random, or that tropical seasonality is inherently complicated. Zooming into the Northern regions of South America, where a heterogeneous coexistence of ecosystem types shapes the land surface, we find that arid regions seem to have low intrinsic seasonal dimensionality compared to more moist regions.

Long-term modes of land surface variability show a rather complex spatial pattern in terms of intrinsic dimensions: Overall, we find values between 6 and 7 (see also the summary in Fig. 5). The values tend to be higher in high altitude and tropical regions, whereas arid regions show low-complexity patterns. Long-term modes of variability in land-surface variables are probably more complex than one would suspect a priori and should be analyzed deeper in the near future.

The analysis shows how a large number of variables can be seamlessly integrated into a rather complex workflow. However,

the results should be interpreted with caution: One criticism of the PCA approach is its tendency to overestimate the correct intrinsic dimensions in the presence of nonlinear dependencies between variables. A second limitation is that the maximum intrinsic dimensions depends on the number of Fourier coefficients used to construct the signals, leading to different theoretical maximum intrinsic dimensions per time scale.





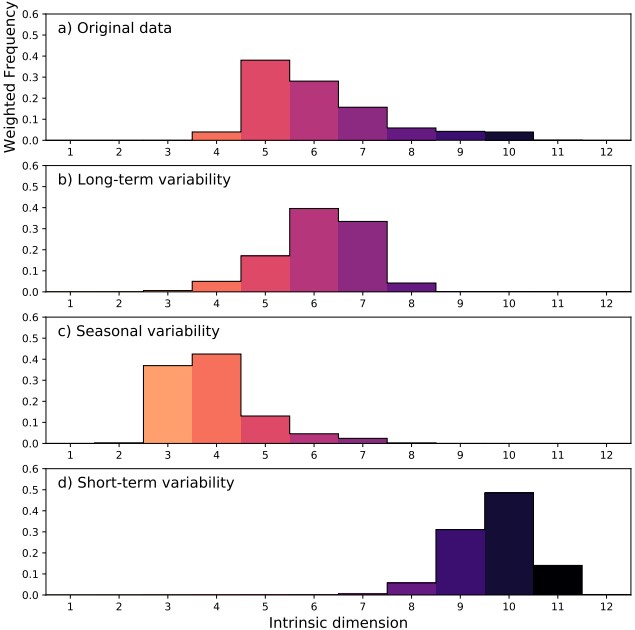

**Figure 5.** Histogram of the intrinsic dimension estimated from 18 land ecosystem variables the Earth system data cube. Highest intrinsic dimension emerges in the short term variability, while the original data are enveloped by the complexity of seasonal and long-term subsignals.

### 4.3 Model-parameter estimation in the ESDL

Another key element in supporting Earth system sciences with the ESDL (and related initiatives) is to enable model development, parametrization, and evaluation. To explore this potential we present a parameter estimation study that aims to reveal sensitivities of ecosystem respiration, $R_{eco}$—the natural release of $CO_2$ by ecosystems to the atmosphere—to fluctuations in temperature. Estimating such sensitivities is key for understanding and modelling the global climate-carbon cycle feedbacks (Kirschbaum, 1995). The following simple model (Davidson and Janssens, 2006) is widely used as a diagnostic description of this process:

$$R_{eco,i} = R_b Q_{10}^{\frac{T_i - T_{ref}}{10}}, \tag{14}$$

where $R_{eco,i}$ is ecosystem respiration at time point $i$ and the parameter $Q_{10}$ is the temperature sensitivity of this process, i.e. the factor by which $R_{eco,i}$ would change by increasing (or decreasing) the temperature $T_i$ by $10°$. An indication of how much respiration we would expect at some given reference temperature $T_{ref}$ is given by the pre-exponential factor $R_b$. Under this model, one can directly estimate the temperature sensitivities from some observed respiration and temperature time series. Technically this is possible and Eq. (15) describes a parameter estimation process as an atomic function,

$$f_{\{time,var\}}^{\{para\},\{time\}} : \mathcal{C}(\{lat, lon, time, var\}) \rightarrow \mathcal{C}(\{lat, lon, par\}), \tag{15}$$





that expects a multivariate time series, and returns a parameter vector. Figure 6a visualizes these estimates, which are comparable to many other examples in the literature (see e.g. Hashimoto et al., 2015) and depict pronounced spatial gradients.

High-latitude ecosystems seem to be particularly sensitive to temperature variability according to such an analysis.

However, it has been shown theoretically (Davidson and Janssens, 2006), experimentally (Sampson et al., 2007), and using model-data fusion (Migliavacca et al., 2015), that the underlying assumption of a constant base rate is not justified. The reason is that the amount of respirable carbon in the ecosystem will certainly vary with the supply, and hence phenology, as well as with respiration limiting factors such as water stress (Reichstein and Beer, 2008). In other words, ignoring the seasonal time

evolution of $R_{\mathrm{b}}$ leads to substantially confounded parameter estimates for $Q_{10}$.

One generic solution to the problem is to exploit the variability of respiratory processes at short-term modes of variability. Specifically, one can apply a timescale dependent parameter estimation (SCAPE; Mahecha et al., 2010b), assuming that $R_{\mathrm{b}}$ varies slowly e.g. on a seasonal and slower timescale. This approach requires some time series decomposition as described in Sec. 4.2. The SCAPE idea requires to rewrite the model, after linearization, such that it allows for a time-varying base rate,

$$\ln R_{\mathrm{eco},i} = \ln R_{\mathrm{b},i} + \frac{T_i - T_{\mathrm{ref}}}{10} \ln Q_{10}. \tag{16}$$

The discrete spectral decomposition into frequency bands of the log-transformed respiration allows to estimate $\ln Q_{10}$ on specific timescales that are independent of phenological state changes (for an in-depth description see Mahecha et al., 2010b, supportig materials). Conceptually, the model estimation process now involves two steps (Eqs. (17) and (18)), a spectral decomposition where we produce a data cube of higher order,

$$f_{\{time\}}^{\{time,freq\}} : \mathcal{C}(\{lat,lon,time,var\}) \rightarrow \mathcal{C}(\{lat,lon,time,var,freq\}) \tag{17}$$

followed by the parameter estimation, which differs from the approach described in Eq.15, as this approach only returns a singular parameter ($Q_{10}$), whereas $\ln R_{\mathrm{b},i}$ now becomes a time series:

$$f_{\{time,var,freq\},}^{\{\},\{time\}} : \mathcal{C}(\{lat,lon,time,var,freq\}) \rightarrow \mathcal{C}(\{lat,lon\}) \times \mathcal{C}(\{lat,lon,time\}) \tag{18}$$

The results of the analysis are shown in Fig. 6b where we find generally a much more homogeneous and better constrained

spatial pattern of $Q_{10}$. As suggested in the site-level analysis by Mahecha et al. (2010b) and later by others (see e.g. Wang et al., 2018) we find a global convergence of the temperature sensitivities. We also find that e.g. semi-arid and savanna-dominated regions clearly show lower apparent $Q_{10}$ (Fig. 6a) compared to the SCAPE approach (Fig. 6b). Discussing these patterns in detail is beyond the scope of this paper, but in general terms these finding are consistent with the expectation that in semiarid ecosystems confounding factors act in the opposing direction (Reichstein and Beer, 2008).

From a more methodological point of view this research application shows that it is well possible to implement a multistep analytic workflow in the ESDL that combines time series analysis and parameter estimation. Once the analysis is implemented, it requires essentially only two sequential atomic functions. The results obtained have the form of a data cube and could be integrated into subsequent analyses. Examples include comparisons with in-situ data, eco-physiological parameter interpretations or assessment of parameter uncertainty in more detail.





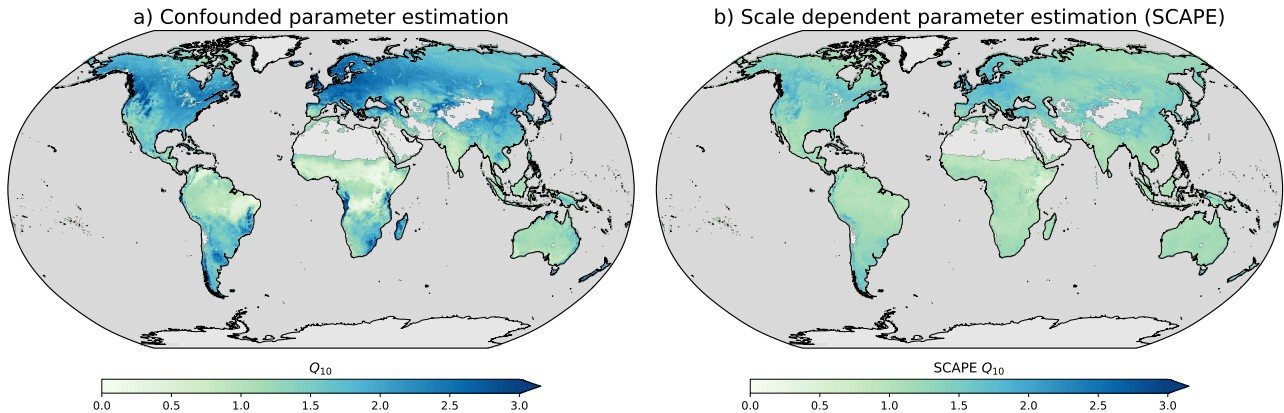

**Figure 6.** Global patterns of locally estimated temperature sensitivities of ecosystem respiration $Q_{10}$, a) via a conventional parameter estimation approach, and b) via a time-scale dependent parameter estimation method. The latter reduce the confounding influence of seasonality and lead to a rather homogeneous map of temperature sensitivity.

## 5 Discussion

In the following, we describe the insights gained during the development of the concept and the implementation of the ESDL, addressing issues arising and critiques expressed during our community consultation processes. We also briefly discuss the ESDL in the light of other developments in the field. Finally, we highlight some challenges ahead and proposed future applications.

### 5.1 Insights and critical perspectives

During a community consultation process across various workshops and summer schools, users expressed confusion about the equitable treatment of data cube dimensions (Sect. 2). Considering that an unordered nominal dimension of "variables" is a dimension as "time" or "latitude" seems counterintuitive at first glance. Yet, the Earth system data cube is sufficiently flexible to allow users to adopt a more classical approach to deal with the Earth system data cube and analyze variable by variable. However, for research examples structured like the second use case (Sect. 4.2), this approach was key as it is allowed to efficiently navigate through the variable dimension. It is obviously irrelevant to algorithms of dimensionality reduction, which dimension is compressed and we could have equally asked the question in time domain or across a spatial dimensions, which relates to the well-known empirical orthogonal functions (EOFs) as used in climate sciences (Storch and Zwiers, 1999). In exploratory approaches of this kind, where there is no prior scientific basis for presupposing where the "information-rich" zones" are in the data cube, a dimension-agnostic approach clearly pays off. We also favour this idea as it is in-line with other approaches discussed in the community. For instance, the "Data Cube Manifesto" (Baumann, 2017) states that *"Datacubes shall treat all axes alike, irrespective of an axis having a spatial, temporal, or other semantics."*, a principle that we have





radically implemented in the `ESDL.jl` Julia package (Sect. 3). The flexibility we gain is that we are, in principle, prepared for comparable cases where one has to deal with e.g. multiple model versions, model ensemble members, or model runs based
on varying initial conditions.

One of the most commonly expressed practical concerns is the choice of an unique data grid. The curation of multiple data streams within such a data cube grid requires that many data have to undergo reformatting and/or remapping. Of course, this can be problematic at times, in particular when data have been produced for a given spatial or temporal resolution and cannot be remapped without violating basic assumptions. For instance keeping mass balances, integrals of flux densities, and global
moments of intensive properties as consistent as possible should always be a priority. However, for the data cube approach implemented here we decided to accept certain simplifications. The availability of a multitude of relevant data to study Earth system dynamics is a key incentive to use the ESDL and goes much beyond many disciplinary domains. But, as we have learned in this discussion, it comes at the price of some pragmatic trade-offs. A fundamental advancement of our approach would be to natively deal with data streams from unequal grids.

Another very practical challenge has to do with living data issues. It is very challenging to maintain a virtual environment up to date: Again, there is a trade-off between the desire to include the latest data streams (ideally even in near real time), and constantly expanding the portfolio according to user needs. The ESDL thus depends on the enduring enthusiasm of the user community and funding agencies to support the idea in this respect and grow steadily into new domains, help us adding data streams, and proactively co-develop the approach.

**5.2   Relation to other initiatives and platforms**

Over the past few years, several initiatives, platforms, and software solutions (Lu et al., 2018; Sudmanns et al., 2019) have emerged based on similar considerations as those motivating the Earth System Data Lab. Some of these platforms and software solutions are explicitly constructed around the idea of data cubes (e.g. Baumann et al., 2016; Lewis et al., 2017; Appel and Pebesma, 2019). Nevertheless, the concept of "data cube" is still not fully consolidated in the Earth system science. Only now
(September 2019) is the Open Geospatial Consortium(OGC) considering establishing standards for data cubes in our branch of science.

Among the other existing initiatives, the Climate Data Store (CDS) of the Copernicus Climate Change Service (https://cds.climate.copernicus.eu/) is conceptually probably the closest one to the ESDL. The CDS was primarily designed as key infrastructure to analyze climate reanalysis data and related variables. These data often require to be analyzed at very high tem-
poral resolutions (e.g. using hourly time-steps). The CDS offers a similar python interface to analyze these data. Likewise, the Google Earth Engine (GEE, https://earthengine.google.com Gorelick et al., 2017) is probably the most widely known platform for implementing global scale analytics. GEE offers access to a wide range of satellite data archives and increasingly also to climate data in their native resolutions. One strength of GEE is the massive computing power offered to the scientist, such that some use cases nicely showcased the power of the infrastructure. The user has a wide range of predefined operators available
that can be used and coupled to build workflows that are particularly suitable for time-series. Another recent development in the field is the Open Data Cube (ODC; https://www.opendatacube.org/; formerly Australian Data Cube; Lewis et al., 2017).



This project was initially designed to offer access to the well processed remote sensing data over Australia with an emphasis on the Landsat archive. In the past years, the ODC technology was used to implement regional data cubes for Colombia (CDCol; Ariza-Porras et al., 2017; Bravo et al., 2017) and Switzerland (SDC: http://www.swissdatacube.org/; Giuliani et al., 2017). The

aim of the open access ODC is also to effectively enable access to time-series data from high-resolution data archives, targeting mainly changes in land surface properties. The ESDL has developed into a conceptually different direction than most of the other initiatives that make it unique:

First, we note that most of the data cube initiatives were motivated by the need to accessing and/or analyze big, e.g. very high resolution data (Lewis et al., 2017; Nativi et al., 2017). Initially, this problem was not in the focus of the ESDL which rather

aimed at downstream data products. Our data cube approach primarily intends to support the joint exploitation of multiple data streams efficiently. This multivariate focus is rarely found as a key design element in the other approaches.

Second, most initiatives intend to preserve the resolutions of the underlying data. The ESDL, instead, is built around singular data cubes that then include variables as an additional dimension. The inevitable trade-off, as discussed above, is the need for a data curation and remapping process prior to the analyses.

Third, there is a wide consensus that data cube technologies need to enable the application of UDFs. However, at this stage, this aspect often appears not to be a priority of other data cube initiatives and, consequently, users are restricted in their analysis by the available tools. In this context we see the strength of the ESDL as it allows for the development of complex workflows and adding arbitrary functionalities efficiently. This is actually one reason why we decided to implement the ESDL in the quite young language of scientific computing Julia (side by side with the more commonly used Python tools).

Taken together, the ESDL has probably conceptually developed (and implemented) the most radical cubing principle following a strict dimension agnostic approach. We envisage that the ESDL front-end could be coupled to a data cube technology as proposed by any of the other initiatives to combine its analytic strength with the efficiencies achieved by others in dealing with high-resolution data streams.

### 5.3 Priorities for future developments

During the development of the ESDL, we identified several several methodological challenges on the one hand and, on the other, application domains that could be addressed. With regard to potentially relevant methodological paths, we can only briefly mention, without claim for completeness, some of the most ardently and widely discussed topics:

– *Machine learning:* Data-driven approaches have always been part of the DNA of Earth system sciences (cf. classical textbooks e.g. Storch and Zwiers, 1999) and classically complement process-driven modelling efforts (Luo et al., 2012).

However, with the rise of modern machine learning new perspectives have emerged (Mjolsness and DeCoste, 2001; Hsieh, 2009). Depending on the purpose we find purely exploratory analysis based on e.g. nonlinear dimensionality reduction (Mahecha et al., 2010a) or predictive techniques (Jung et al., 2009) being transferred from computer sciences to the Earth system sciences. Today, deep learning is on everybody's lips and could mark one step forward in Earth system science (Karpatne et al., 2018; Shen et al., 2018; Bergen et al., 2019; Reichstein et al., 2019). Through providing an



easy access to relevant data streams, the Earth system data cube idea may attract further researchers from data sciences
into the field. It furthermore provides the perfect platform for studying complex tasks such as detecting multidimen-
sional extreme events (Flach et al., 2017), characterization of information content and dependencies in the data with
information-theoretic measures (Sippel et al., 2016), or causal inference (Runge et al., 2019; Pearl, 2009; Peters et al.,
2017; Christiansen and Peters, 2018). We believe that the clear and easy-to-use interface of the ESDL renders it well
suited for being part of machine learning challenges such as the ones organized by kaggle or during premier conferences
of the field.

– *Spatial interactions:* For interpreting the interactions and mechanisms of the land and ocean, or land and atmosphere
that involve lateral transport, the ESDL would require more developments. Statistical approaches like spatial network
analyses (e.g. Donges et al., 2009; Boers et al., 2019), or process oriented ideas like explicit moisture transport (e.g.
Wang-Erlandsson et al., 2018) would be very valuable to be explored, but would require a substantial rethinking of the
the actual implementation in order to achieve high performances.

– *Model evaluation and benchmarking:* Our third use-case (Sect. 4.3) illustrates the suitability of the ESDL for parameter
estimation and model-evaluation purposes. Today, typical model evaluation frameworks in the Earth system sciences
prepare predefined benchmark metrics on some reference data sets (Luo et al., 2012). Prominent examples are the
benchmarking tools awaiting the CMIP6 model suites (Eyring et al., 2019). However, these model-evaluation frame-
works typically do not give the user the full flexibility to apply some user-defined metrics to the model ensemble under
scrutiny. We believe that mapping UDFs on such big Earth system model output could greatly benefit the development of
novel evaluation metrics in the near future. Building data cubes from multi-model ensembles would be straightforward,
as different models or ensembles would simply lead to one additional dimension in our setup.

In terms of application domains we see high potential in the following areas:

– *Human-environment interactions:* Addressing the complexities of "human-environmental interactions" (Schimel et al.,
2015) is a particular challenge. Making the ESDL fit for this purpose would require integrating a variety of (at least)
spatially explicit population estimates (Doxsey-Whitfield et al., 2015) and socioeconomic data Smits and Permanyer
(2019). The latter represent a fundamentally novel development that has great potential for understanding e.g. dynamics
of disasters impacts (Guha-Sapir and Checchi, 2018), among other issues. In fact this integration is a grand challenge
ahead (Mahecha et al., 2019), but not out of reach for the ESDL.

– *Biodiversity research:* Another question of high societal relevance is to understand how patterns of biodiversity affect
ecosystem functioning (Emmett Duffy et al., 2017; García-Palacios et al., 2018). In the light of a global decline in species
richness (cf. latest global reports https://www.ipbes.net/), this question is of uttermost importance. The ESDL is only
partly fit for this purpose, as it would require the ingestion of a wide range of essential biodiversity variables (Pereira
et al., 2013; Skidmore et al., 2015), beyond the ones we have already available. But still, the ESDL is conceptually
prepared to deal with these challenges (compare e.g. the demands described in Hardisty et al., 2019) and would be



particularly suitable for relating biodiversity patters to the so-called ecosystem function properties (Reichstein et al., 2014; Musavi et al., 2015). In fact, in the regional application of the ESDL we have focused on Colombia and its wider

region to explore linkages of this kind relying on remote sensing derived variables that are relevant for this context.

– *Oceanic sciences:* Extending the ESDL for ocean data is desired and conceptually possible. Surface parameters, e.g. on phytoplankton phenology derived from remote sensing (Racault et al., 2012), can be treated analogously to terrestrial surface parameters. Other dynamics, e.g. the analysis and exploration of ocean-land coupling mechanisms, ocean-atmosphere interactions, and land-atmosphere interactions triggered by ocean circulation dynamics could in principle be

facilitated via the ESDL but require to either vertical or lateral dynamics.

– *Solid Earth:* The step towards global, fully data informed, model data is also made in geophysics. For instance, recently Afonso et al. (2019) used an inversion approach to develop a 3D model that fully describes multiple parameters in the Earth interior, including e.g. crustal and lithospheric thickness, average crustal density, and a depth-dependent density of the lithospheric mantle, among other variables. They proposed a tool allowing for inspecting the data interactively

at a spatial resolution of $2° \times 2°$ grid in different depth. Clearly, in this case other dimensions are relevant as in our implementation, but the principle is the same and, in fact, treated in a very similar manner. Future model-data assimilation approaches of this kind could be performed in the context of the ESDL, as well as the aforementioned machine learning for the solid Earth (Bergen et al., 2019).

In summary, we have demonstrated that the ESDL is a flexible and generic framework that can allow various different

communities to explore and analyse large amounts of gridded data efficiently. Thinking about the potential paths ahead, the ESDL could become a valuable tool in various fields of Earth system sciences, biodiversity research, computer sciences and other branches of science. The widespread social and political uptake of the concept of planetary boundaries (Rockström et al., 2009; Steffen et al., 2015) underlines the global demand for better quantified process understanding of environmental risks and resource bottlenecks based on empirical evidence. Along these lines, the ESDL concept could be used to address some

of the most pressing global challenges. For example, it could become an interface for direct interaction with ECVs, global climate projections and EBVs. Such an interactive interface would allow a much broader community to better understand the data underlying the global assessment reports of the IPCC (Pachauri et al., 2014) and IPBES (Diaz et al., 2019). If coupled to some visual interfaces, the ESDL could also be used by a broader community, enhancing education, communication and decision making process, contributing to knowledge democratization about a deeper understanding of the complex and dynamic

interactions in the Earth system.





# 6    Conclusions

Exploiting the synergistic potential of multiple data streams in the Earth sciences beyond disciplinary boundaries requires a common framework to treat multiple data dimensions, such as for instance spatial, temporal, variable, frequency and other grids, alike. This idea leads to a data cube concept that opens novel avenues to efficiently deal with data in the Earth system

sciences. In this paper, we have formalized the concept of data cubes and described a way to operate on them. The outlined dimension-agnostic approach is implemented in the Earth System Data Lab that enables users applying a wide range of functions to all thinkable combinations of dimension. We believe that this idea can dramatically reduce the barrier to exploit Earth system data and serves multiple research purposes. The ESDL complements a range of emerging initiatives that differ in architectures and specific purposes. However, the ESDL is probably the most radical data cubing approach, offering novel

opportunities for cross-community data-intensive exploration of contemporary global environmental changes. Future developments in related branches of science and latest methodological developments need to be considered and addressed soon. However, already at its actual state of implementation, the ESDL promises to contribute to the deeper understanding and more effective implementation of policy-relevant concepts such as the planetary boundaries, essential variables in different subsystems of the Earth, and global assessment reports. We see a particularly high future potential in dealing with large scale model

ensembles for evaluation tasks, or coupling the developed front-end with a data handling strategy that could deal with new generations of satellite remote sensing data with their constantly increasing spatial, temporal, and spectral resolutions.

*Author contributions.* Conceptual development: M.D.M., F.G. and M.R.; Implementation of the `ESDL.jl` package in in the Julia language: F.G.; Implementation of the ESDL: N.F., F.G., M.D.M., G.B.; Notation: R.C., J.P., M.D.M., F.G., G.K., P.P.; Data: D.G.M., M.J.; Paper writing: M.D.M. wrote the manuscript with substantial input from F.G., R.C., J.P. and comments from all co-authors.

*Competing interests.* The authors declare no competing interests.

*Code availability.* All code nessecary to build and analyze the ESDL is available from https://github.com/esa-esdl. The case studies presented in Sect. can be fully reproduced from https://github.com/esa-esdl/ESDLPaperCode.jl.

*Data availability.* All data are available via earthsystemdatalab.net or from the original data providers as indicated in the manuscript.

*Competing interests.* The authors declare no competing interests.



*Acknowledgements.* This paper was funded by the European Space Agency (ESA) via the "Earth System Data Lab" (ESDL) project. All authors thank the ESA and the Integrated Land Ecosystem Atmosphere Processes Study (iLEAPS), Global Research Project for constant support. The implementation of the regional Earth data cube for Colombia was done under the project "Champion user phase; Supporting the Colombia BON in GEO BON" with the ESDL project. The original idea emerged at the iLEAPS–ESA–MPG funded workshop in Frascati 2011 (Mahecha et al., 2011). We thank everyone participating in the various workshops, summer schools, and early adopters, providing

invaluable feedback on the ESDL. We thank everyone who made data freely available such that they could be used in this project. Special thanks to Eleanor Blyth, Carsten Brockmann, Garry Hayman, Toby R. Marthews, and Uli Weber for constant support and critical feedback. R.C. and J.P. were supported by a research grant (18968) from VILLUM FONDEN. G.C.V. was supported by the ERC under the ERC-COG-2014 SEDAL (grant agreement 647423); .G.M. by the ERC under grant agreement no. 715254 (DRY–2–DRY). J.F.D. was supported by the Stordalen Foundation (via the Planetary Boundary Research Network) and the ERC via the ERC advanced grant project ERA (Earth

resilience in the Anthropocene).



**Appendix A: Data streams in the Earth system data lab**

In the following we give an overview of the actually available variables in the Earth system data lab. The list is constantly being updated.

Table A1: Data streams in the current implementation of the ESDL.

| Domain | Variable | Short | Coverage | Description | References |
|---|---|---|---|---|---|
| Atmosphere | 2 metre temperature | $t2m$ | 2001–2011 | The air temperature at 2 m data ($[T_{2m}] =$ K) are part of the ERA-Interim reanalysis product, and therefore produced by data assimilation techniques in combination with a forecast model. The original spatial sampling (T255 spectral resolution) approximates to 80 km and the original temporal sampling is 6 hours for analyses and at 3 hours for forecasts. | Dee et al. (2011) |
| Atmosphere | Aerosol Optical Thickness at 550 nm | $AOD550$ | 2002–2012 | The ESA CCI Aerosol Optical Thickness (Depth) data sets were created by using algorithms, which were developed in the ESA aerosol_cci project. The data used here were created from AATSR measurements (ENVISAT mission) using the ...... algorithm and represent total column AOD at the specified wavelength. Horizontal resolution of the daily data is 1 degree x 1 degree on a global grid. | Holzer-Popp et al. (2013) |





Table A1: Data streams in the current implementation of the ESDL.

| Domain | Variable | Short | Coverage | Description | References |
|---|---|---|---|---|---|
| Atmosphere | Aerosol Optical Thickness at 555 nm | $AOD555$ | 2002–2012 | The ESA CCI Aerosol Optical Thickness (Depth) data sets were created by using algorithms, which were developed in the ESA aerosol_CCI project. The data used here were created from AATSR measurements (ENVISAT mission) using the ...... algorithm and represent total column AOD at the specified wavelength. Horizontal resolution of the daily data is 1 degree x 1 degree on a global grid. | Holzer-Popp et al. (2013) |
| Atmosphere | Aerosol Optical Thickness at 659 nm | $AOD659$ | 2002–2012 | The ESA CCI Aerosol Optical Thickness (Depth) data sets were created by using algorithms, which were developed in the ESA aerosol_cci project. The data used here were created from AATSR measurements (ENVISAT mission) using the ...... algorithm and represent total column AOD at the specified wavelength. Horizontal resolution of the daily data is 1 degree x 1 degree on a global grid. | Holzer-Popp et al. (2013) |
| Atmosphere | Aerosol Optical Thickness at 865 nm | $AOD865$ | 2002–2012 | The ESA CCI Aerosol Optical Thickness (Depth) data sets were created by using algorithms, which were developed in the ESA aerosol_cci project. The data used here were created from AATSR measurements (ENVISAT mission) using the ...... algorithm and represent total column AOD at the specified wavelength. Horizontal resolution of the daily data is 1 degree x 1 degree on a global grid. | Holzer-Popp et al. (2013) |



Table A1: Data streams in the current implementation of the ESDL.

| Domain | Variable | Short | Coverage | Description | References |
|--------|----------|-------|----------|-------------|------------|
| Atmosphere | Aerosol Optical Thickness at 1610 nm | $AOD1610$ | 2002–2012 | The ESA CCI Aerosol Optical Thickness (Depth) data sets were created by using algorithms, which were developed in the ESA aerosol_cci project. The data used here were created from AATSR measurements (ENVISAT mission) using the ...... algorithm and represent total column AOD at the specified wavelength. Horizontal resolution of the daily data is 1 degree x 1 degree on a global grid. | Holzer-Popp et al. (2013) |
| Biosphere | Gross Primary Productivity | $GPP$ | 2001–2012 | By training an ensemble of machine learning algorithms with eddy covariance data from FLUXNET and satellite observations in a cross-validation approach, regressions from these observations to different kinds of carbon and energy fluxes were established and used to generate datasets with a spatial resolution of 5 arc-minutes and a temporal resolution of 8 days. The GPP resembles the total carbon release of the ecosystem through respiration and is expressed in the unit gC $m^{-2}$ $day^{-1}$. | Tramontana et al. (2016) |





Table A1: Data streams in the current implementation of the ESDL.

| Domain | Variable | Short | Coverage | Description | References |
|---|---|---|---|---|---|
| Biosphere | Net Ecosystem Exchange | $NEE$ | 2001–2012 | By training an ensemble of machine learning algorithms with eddy covariance data from FLUXNET and satellite observations in a cross-validation approach, regressions from these observations to different kinds of carbon and energy fluxes were established and used to generate datasets with a spatial resolution of 5 arc-minutes and a temporal resolution of 8 days. The NEE resembles the net carbon exchange between the ecosystem and the atmosphere and is expressed in the unit gC m$^{-2}$ day$^{-1}$. | Tramontana et al. (2016) |
| Land | Latent Energy | $LE$ | 2001–2012 | By training an ensemble of machine learning algorithms with eddy covariance data from FLUXNET and satellite observations in a cross-validation approach, regressions from these observations to different kinds of carbon and energy fluxes were established and used to generate datasets with a spatial resolution of 5 arc-minutes and a temporal resolution of 8 days. The LE resembles the latent heat flux from the surface and is expressed in the unit W m$^{-2}$. | Tramontana et al. (2016) |



Table A1: Data streams in the current implementation of the ESDL.

| Domain | Variable | Short | Coverage | Description | References |
|---|---|---|---|---|---|
| Land | Sensible Heat | $H$ | 2001–2012 | By training an ensemble of machine learning algorithms with eddy covariance data from FLUXNET and satellite observations in a cross-validation approach, regressions from these observations to different kinds of carbon and energy fluxes were established and used to generate data sets with a spatial resolution of 5 arc-minutes and a temporal resolution of 8 days. The H resembles the sensible heat flux from the surface and is expressed in the unit $W\,m^{-2}$. | Tramontana et al. (2016) |
| Land | Monthly Burnt Area | $BurntArea$ | 1995–2014 | This data set was taken from the fourth generation of the Global Fire Emissions Database (GFED4). It was created as a combination of data from infrared sensor satellite observations and resembles the estimated monthly burnt area in hectares. The spatial resolution of this data set is 0.25°. Small fires were exempt in the production of the data. | Giglio et al. (2013) |
| Land | Carbon dioxide emissions due to natural fires expressed as carbon flux | $Emission$ | 2001–2010 | This data set was taken from the fourth generation of the Global Fire Emissions Database (GFED4). It was created by applying a model based on the Carnegie-Ames-Stanford Approach (CASA) to the burnt area estimates and has the same temporal (monthly) and spatial (0.25°) resolution as the monthly burnt area data set and expresses the carbon dioxide emissions of natural fires as a carbon flux ($gC\,m^{-2}\,day^{-1}$). Small fires were included in this approach. | Giglio et al. (2013); van der Werf et al. (2017) |



Table A1: Data streams in the current implementation of the ESDL.

| Domain | Variable | Short | Coverage | Description | References |
|---|---|---|---|---|---|
| Land | Evaporation | $E$ | 2001–2011 | The GLEAM data sets are created by using a set of algorithms, input forcing data sets from reanalyses, optical and microwave satellites and other merged sources. The model itself consists of four modules: potential evaporation (Priestley and Taylor equation), interception (Gash analytical model), soil (mulit-layer soil model + data assimilation) and stress (semi-empirical). The data are sampled on a graticule of 0.25° and have a daily temporal coverage. | Martens et al. (2017); Miralles et al. (2011) |
| Land | Evaporative Stress Factor | $S$ | 2001–2011 | The GLEAM data sets are created by using a set of algorithms, input forcing data sets from reanalyses, optical and microwave satellites and other merged sources. The model itself consists of four modules: potential evaporation (Priestley and Taylor equation), interception (Gash analytical model), soil (mulit-layer soil model + data assimilation) and stress (semi-empirical). The data are sampled on a graticule of 0.25° and have a daily temporal coverage. | Martens et al. (2017); Miralles et al. (2011) |



Table A1: Data streams in the current implementation of the ESDL.

| Domain | Variable | Short | Coverage | Description | References |
|---|---|---|---|---|---|
| Land | Potential Evaporation | $Ep$ | 2001–2011 | The GLEAM data sets are created by using a set of algorithms, input forcing data sets from reanalyses, optical and microwave satellites and other merged sources. The model itself consists of four modules: potential evaporation (Priestley and Taylor equation), interception (Gash analytical model), soil (mulit-layer soil model + data assimilation) and stress (semi-empirical). The data are sampled on a graticule of 0.25° and have a daily temporal coverage. | Martens et al. (2017); Miralles et al. (2011) |
| Land | Interception Loss | $Ei$ | 2001–2011 | The GLEAM data sets are created by using a set of algorithms, input forcing data sets from reanalyses, optical and microwave satellites and other merged sources. The model itself consists of four modules: potential evaporation (Priestley and Taylor equation), interception (Gash analytical model), soil (mulit-layer soil model + data assimilation) and stress (semi-empirical). The data are sampled on a graticule of 0.25° and have a daily temporal coverage. | Martens et al. (2017); Miralles et al. (2011) |



Table A1: Data streams in the current implementation of the ESDL.

| Domain | Variable | Short | Coverage | Description | References |
|---|---|---|---|---|---|
| Land | Root-Zone Soil Moisture | $SMroot$ | 2001–2011 | The GLEAM data sets are created by using a set of algorithms, input forcing data sets from reanalyses, optical and microwave satellites and other merged sources. The model itself consists of four modules: potential evaporation (Priestley and Taylor equation), interception (Gash analytical model), soil (mulit-layer soil model + data assimilation) and stress (semi-empirical). The data are sampled on a graticule of 0.25° and have a daily temporal coverage. | Martens et al. (2017); Miralles et al. (2011) |
| Land | Surface Soil Moisture | $SMsurf$ | 2001–2011 | The GLEAM data sets are created by using a set of algorithms, input forcing data sets from reanalyses, optical and microwave satellites and other merged sources. The model itself consists of four modules: potential evaporation (Priestley and Taylor equation), interception (Gash analytical model), soil (mulit-layer soil model + data assimilation) and stress (semi-empirical). The data are sampled on a graticule of 0.25° and have a daily temporal coverage. | Martens et al. (2017); Miralles et al. (2011) |





Table A1: Data streams in the current implementation of the ESDL.

| Domain | Variable | Short | Coverage | Description | References |
|---|---|---|---|---|---|
| Land | Bare Soil Evaporation | $Eb$ | 2001–2011 | The GLEAM data sets are created by using a set of algorithms, input forcing data sets from reanalyses, optical and microwave satellites and other merged sources. The model itself consists of four modules: potential evaporation (Priestley and Taylor equation), interception (Gash analytical model), soil (mulit-layer soil model + data assimilation) and stress (semi-empirical). The data are sampled on a graticule of 0.25° and have a daily temporal coverage. | Martens et al. (2017); Miralles et al. (2011) |
| Land | Snow Sublimation | $Es$ | 2001–2011 | The GLEAM data sets are created by using a set of algorithms, input forcing data sets from reanalyses, optical and microwave satellites and other merged sources. The model itself consists of four modules: potential evaporation (Priestley and Taylor equation), interception (Gash analytical model), soil (mulit-layer soil model + data assimilation) and stress (semi-empirical). The data are sampled on a graticule of 0.25° and have a daily temporal coverage. | Martens et al. (2017); Miralles et al. (2011) |



Table A1: Data streams in the current implementation of the ESDL.

| Domain | Variable | Short | Coverage | Description | References |
|---|---|---|---|---|---|
| Land | Transpiration | $Et$ | 2001–2011 | The GLEAM data sets are created by using a set of algorithms, input forcing data sets from reanalyses, optical and microwave satellite sensors and other merged sources. The model itself consists of four modules: potential evaporation (Priestley and Taylor equation), interception (Gash analytical model), soil (mulit-layer soil model + data assimilation) and stress (semi-empirical). The data are sampled on a graticule of 0.25° and have a daily temporal coverage. | Martens et al. (2017); Miralles et al. (2011) |
| Land | Open-water Evaporation | $Ew$ | 2001–2011 | The GLEAM data sets are created by using a set of algorithms, input forcing data sets from reanalyses, optical and microwave satellite sensors and other merged sources. The model itself consists of four modules: potential evaporation (Priestley and Taylor equation), interception (Gash analytical model), soil (mulit-layer soil model + data assimilation) and stress (semi-empirical). The data are sampled on a graticule of 0.25° and have a daily temporal coverage. | Martens et al. (2017); Miralles et al. (2011) |
| Land | White Sky Albedo for Visible Wavelengths | $BHR\_VIS$ | 1998–2012 | White sky albedo, also known as bi-hemispherical reflectance (only diffuse illumination), estimated from satellite radiometer data. The spatial resolution of this product is 1 km with a temporal sampling of 8 days. | Lewis et al. (2012) |





Table A1: Data streams in the current implementation of the ESDL.

| Domain | Variable | Short | Coverage | Description | References |
|---|---|---|---|---|---|
| Land | Black Sky Albedo for Visible Wavelengths | $DHR\_VIS$ | 1998–2012 | Black sky albedo, also known as directional-hemispherical reflectance (only direct illumination), estimated from satellite radiometer data. The spatial resolution of this product is 1 km with a temporal sampling of 8 days. | Lewis et al. (2012) |
| Water | Fractional Snow Cover | $MFSC$ | 2003–2013 | Global fractional snow cover product using mainly satellite infrared radiometer data (ATSR-2, AATSR). Glaciers, continental ice shields and snow on ice are exempt from the data. Values stand for the percentage of the area of a grid cell covered by snow integrated over time (daily, weekly or monthly). The spatial resolution is 1 km. | Luojus et al. (2010); Metsämäki et al. (2015) |
| Water | Snow Water Equivalent | $SWE$ | 1980–2012 | Snow water equivalent product covering the northern hemisphere (35°N–85°N), created by using microwave sensor data (SMMR, SSM/I, SSMIS). Glaciers, continental ice shields and mountaineous regions are exempt from the data. Values stand for the water equivalent of snow per grid cell in millimetres aggregated over time (daily, weekly or monthly). The weekly data is produced by giving every day the mean value of a sliding window (-6 days). The monthly data is given as the weekly mean and maximum per calendar month. The spatial resolution is approximately 25 km. | Luojus et al. (2010) |





Table A1: Data streams in the current implementation of the ESDL.

| Domain | Variable | Short | Coverage | Description | References |
|---|---|---|---|---|---|
| Land | Land Surface Temperature | *LST* | 2002–2011 | The GlobTemperature Land Surface Temperature product used here is a product of a satellite infrared radiometer (AATSR). It has global coverage with a spatial sampling of 0.05° and consists of 2 measurement averages (day and night). The values are an approximation of the average land surface temperature per grid cell in K. It is an improved version of the ESA AATSR data set (UOL_LST_3P, v2.1). | Ghent (2012) |
| Atmosphere | Total Column Water Vapour | *TCWV* | 1996–2008 | The TCWV product was derived through combination of various satellite spectrometer and microwave sensor data sets. It resembles the total mass of water contained in a column of air from the surface to 200 hPa. The unit is kg m$^{-2}$, the spatial sampling is 0.5° and the data is provided as daily composites. From 1996–2002 including, the data consists of weekly/monthly means. | Schröder et al. (2012); Schneider et al. (2013) |
| Atmosphere | Precipitation | *Precip* | 1980–2015 | The Global Precipitation Climatology Project (GPCP) | Adler et al. (2003); Huffman et al. (2009) |





Table A1: Data streams in the current implementation of the ESDL.

| Domain | Variable | Short | Coverage | Description | References |
|---|---|---|---|---|---|
| Atmosphere | Mean total ozone column | $Ozone$ | 1996–2011 | The total ozone column data from the Ozone CCI project is derived from GOME spectrometer acquisitions. For the ESDL, Level 2 data have been used. They are given in Dobson units (DU) and have a spatial resolution of 320 km x 40 km. The temporal resolution depends on the latitude, with the longest revisit time being 3 days at the equator. | Van Roozendael et al. (2012); Lerot et al. (2014) |
| Land | Fraction of Absorbed Photosynthetically Active Radiation | $fAPAR\_TIP$ | 1982–2016 | The fAPAR, describing the amount and productivity of vegetation, was derived by using a Two Stream Inversion Package (TIP) method based on the Two-stream model developed by Pinty et al. (2006). The product is delivered in two spatial resolutions (0.05° and 0.5°) and with a daily temporal coverage. | Disney et al. (2016); Blessing and Löw (2017) |
| Land | Leaf Area Index | $LAI$ | 1982–2016 | The LAI, defined as half the total canopy area per unit ground area ($m^2\ m^{-2}$), was derived by using a Two Stream Inversion Package (TIP) method based on the Two-stream model developed by Pinty et al. (2006). The product is delivered in two spatial resolutions (0.05° and 0.5°) and with a daily temporal coverage. | Disney et al. (2016); Blessing and Löw (2017) |





Table A1: Data streams in the current implementation of the ESDL.

| Domain | Variable | Short | Coverage | Description | References |
|---|---|---|---|---|---|
| Land | White Sky Albedo for Visible Wavelengths from AVHRR | $BHR\_VIS$ | 1982–2016 | White sky albedo, also known as bi-hemispherical reflectance (only diffuse illumination), estimated from satellite radiometer data. This data set extends the GlobAlbedo data by using additional input data sources (AVHRR, geostationary satellites). The product is delivered in two spatial resolutions (0.05° and 0.5°) and with a daily temporal coverage. | Lewis et al. (2012); Danne et al. (2017) |
| Land | Black Sky Albedo for Visible Wavelengths from AVHRR | $DHR\_VIS$ | 1982–2016 | Black sky albedo, also known as directional-hemispherical reflectance (only direct illumination), estimated from satellite radiometer data. This data set extends the GlobAlbedo data by using additional input data sources (AVHRR, geostationary satellites). The product is delivered in two spatial resolutions (0.05° and 0.5°) and with a daily temporal coverage. | Lewis et al. (2012); Danne et al. (2017) |
| Land | Fraction of Absorbed Photosynthetically Active Radiation from AVHRR | $fAPAR\_AVHRR$ | 1982–2006 | The AVHRR derived fAPAR, describing the amount and productivity of vegetation, was derived from AVHRR black sky albedo data. The product is delivered in two spatial resolutions (0.05° and 0.5°) and with a daily temporal coverage. | Gobron et al. (2017) |





Table A1: Data streams in the current implementation of the ESDL.

| Domain | Variable | Short | Coverage | Description | References |
|--------|----------|-------|----------|-------------|------------|
| Land | Soil Moisture | $SM$ | 1978–2017 | The ESA CCI Soil Moisture data combine various active and passive microwave sensors into a homogenised product. It represents the soil water content in the upper 5 cm of the soil. Produced at a spatial sampling of 0.25° and a temporal sampling of one day. Gaps in periods of snow cover or frozen conditions, and in areas with very dense vegetation. | Liu et al. (2012); Dorigo et al. (2017); Gruber et al. (2017) |



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
