# Peer review of "Earth system data cubes unravel global multivariate dynamics"

_Earth System Dynamics, 2019_

## Referee Comment (RC1) · Anonymous Referee #1 · 12 Nov 2019

Dear thank you for giving me the opportunity to review this paper. The paper presents a new datacube approach, specifically designed to study Earth System science dynamics. The paper is well-structured, of high-quality and of interest for a wider Earth System Dynamics audience.

I accept the publication of this paper with minor revisions. Please find my comments for minor revision and a list of typing errors to be addressed attached.

Please also note the supplement to this comment:
https://www.earth-syst-dynam-discuss.net/esd-2019-62/esd-2019-62-RC1-supplement.pdf

[Figure]

**Supplement:**

**ESD-review - Earth system data cubes unravel global multivariate dynamics**

The authors present a new data cube approach, called Earth System Data Cube, where multiple spatiotemporal data streams are treated as one singular, very high-dimensional data stream.

This paper is of high quality and clearly outlines the authors' reasoning of implementing the ESDC in the way it is implemented and the advantages it brings to Earth System dynamics studies.

The ESDC datacube approach brings different data streams to a common grid, which is beneficial for specific scientific Earth System studies. Common operations, as e.g. resampling and bringing the data to the same spatial and temporal resolution, are just taken care of and users can focus more on science.
Yet, the unified grid from the beginning might be a limitation for other application areas.

**Some comments and areas for minor revision:**

What is not so clear for me so far is the overall implementation strategy of the ESDC. I understand that I can now use the ESDC with the datasets as outlined in the paper appendix. Is there the option that users can extend the ESDC data list? Or will it be similar to the OpenDataCube concept where multiple implementations of the ESDC concept can be set up with different data? Where is the current ESDC data hosted? And how long does the data preparation take? I suggest to bring in these additional aspects.

What are the bottlenecks in gathering the data? I understand that in order to implement the ESDC, data has to be moved from the respective data repositories. Is this correct? Would be good to elaborate on this aspect as well.

It is without doubt that Julia is a powerful and efficient programming language. However, the fact that the ESDC package has been developed in Julia could be a restrictive factor in the uptake, as I argue that most potential users of the ESDC use Python or R and might be less motivated to take up a new programming language. Are there plans to extend it to Python or R?

The authors argue that the ESDL is closest to the Climate Data Store. In my current understanding of both data systems, I would disagree. Do the authors talk about the Climate Data Store, which is primarily a data dissemination system, or the Climate Data Store toolbox, which is the processing editor on top of the CDS?
It would be necessary to elaborate more on what aspects both systems are close and why the authors come to this assessment. It is also important that the authors differentiate between the CDS and the CDS toolbox.

**Some additional typing errors discovered:**

page 2 line 21: remove 'the' many key processes
page 4 line 109: remove 'are' … variable only can be considered
page 8 line 174: form instead of from
page 11 line 230: remove 'computing' … scientific programming language
page 12 line 284 don't understand the last part of the sentence 'data speed up data access
page 12 line 287 add 'be' 'would be to treat every …
page 13 line 303 behaviour is doubled 'long-term system behaviour'
page 17 line 370 'confirm' instead of confirms
page 19 line 428 'these findings' instead of finding
page 21 line 456 a unique data grid instead of an unique datagrid
page 21 line 475 rephrase sentence, hard to understand
page 22 line 510 'several' is double

---

## Short Comment (SC1) · 12 Nov 2019

**Discussion of *Earth system data cubes unravel global multivariate dynamics*, by Miguel Mahecha et al.**

Edzer Pebesma,* Marius Appel*

Nov 12, 2019

We congratulate the authors of this paper with a very nice contribution describing the ESDC and its application, an activity that has involved a substantial conceptual development as well as implementation work, that has triggered a number of scientific papers already, and that now helps furthering the discussion what spatiotemporal data cubes are. We are mostly interested in having this latter discussion.

**1 Pre-grid all the data, or do this on-the-fly?**

The ESDC pre-grids all the data, and (l456) "One of the most commonly expressed practical concerns is the choice of a unique data grid". Given that an ESDC is defined on a single grid, but integrates a lot of different datasets, each of these datasets have to be forged into the target grid. This involves resampling, statistical downscaling, interpolation and/or aggregation of data both in space and time, as well as handling coordinate reference systems and possibly different calendars (Gregorian, 365day/noleap, 360 day) into one. It is unclear whether the ESDC can do all this, and also if it can give an idea about the errors introduced by doing so, or whether it can give users otherwise recommendations what a good grid is?

Other systems, e.g. Google Earth Engine or Sentinel Hub work from the raw data (which can be in many different coordinate reference systems, e.g. Sentinel-2 distributed over 120 UTM zones), and create user-defined data cubes on the fly. This has the advantage that low-resolution computations can be done relatively fast and interactively, which helps data exploration and model development, and that the effect of different target resolutions can be easily evaluated. With the ESDC, once data are cubed, users no longer have the possibility to go back to the original data.

We think this issue is important, and missed a discussion on this matter in the paper. Given that large, operational systems exist that do not store pre-
* * *
*Institute for Geoinformatics, University of Münster, Germany

cubed datasets but that work with data cube *views* (Pebesma et al., 2019) we believe that this may be at least a viable option.

**2 Is latitude the same as time?**

In Line 598, the paper states that "The ESDL is probably the most radical data cubing approach", and refers to some grey literature that also claims that data cubes should treat all dimensions identically, irrespective their semantics. We believe however that space and time are inherently different, and need different treatment. The example (e.g. fig 3) shows that all longitudes are aggregated to give profiles per latitude and time, but we feel that this is a rather contrived example; in general, any transect in space could be a good candidate for reducing space to one dimension, and this would require mapping onto that transect; it is not so likely that you would want to do that on an arbitrary transect in the two-dimensional space defined by e.g. (longitude, time). In general, a large number of functions applied to space operate on both spatial dimensions (e.g., polygonal crop), and time series models are of a very different kind and are rarely applied to single spatial dimensions. The fact that you *can* do this is nice, but we do not believe it more of a big selling point than an opportunity for users to shoot themselves in the foot.

**3 Is a lat/long grid the only way we can cube the Earth?**

No, it isn't, and many datasets use other global grids, discrete global grids, or collections of grids (Equi7, or UTM zones). All this has reasons, and the Earth will never be flat so the problem will remain. A discussion on generalizing the ESDC to other grids, or collections of these, would be welcome.

**4 Vector data cubes are missing**

In the representation of the ESDC, the implicit assumption is being made that data cubes correspond to spatial raster data. This is not the case: space can be a one-dimensional set of feature geometries (e.g. points, or polygons). One of the most requested feature of data cubes is to retrieve all the cube information at a given set of points, or aggregated over a given set of polygons. This leads naturally to vector data cubes. Can ESDC answer such queries, or do the authors consider this to not be data cubes?

**5 Where is support?**

Spatial grids may refer to a collection of points on a regular grid, or to a collection of grid cells as if they were small square polygons. In the latter case,

properties can be either continuous over a grid cell (e.g., land use, soil type, geology) or may be aggregated values over the grid cell (e.g., the total amount of carbon in the grid cell, or its maximum elevation). Similarly can time be conceived as a set of time instances or intervals, with the two interval interpretations. Does the ESDC take care of some of these options, or is this all assumed to be remembered by the users?

**6 Code and reproducibility**

We are happy to learn that the software is open source, and can be used by others. Have the authors heard success stories of others installing and using the software? A few small code examples in the paper to get a taste for how simple queries to the ESDC look like would have worked well, and could be encourage those hungry for trying it out. We did not try out the Julia script but hope that other reviewers will report whether they did, and whether they were successful in reproducing the results shown in the paper.

**7 Dropping dimensions**

If a dimension has only one value, it is dropped; we can see this is useful, but it also drops the information of the dimension (e.g. the species name, or the elevation value). We see a lot of 4-dimensional NetCDF files in the wild having 1 dimension with one values, to specify the value for that dimension, which seems all but useless. You need to be able to drop it, but can the ESDC also not drop it, e.g. so that sub-cubes can be meaningfully combined along the otherwise dropped dimension?

**8 The data model**

Having a data model that maps from dimensions to $\mathbb{R}$ and NA is great; a similar approach was adopted in Appel and Pebesma (2019). For end users it also means there is no way to properly handle logical (TRUE/FALSE or NA), categorical, or e.g. time variables. Of course 8-byte doubles can encode anything, but everyone who has tried to use them for encoding categories knows the nightmare. Do end users need that? Some sort of a discussion on this issue would be welcome.

**9 Other issues**

1. Can the ESDC cope with irregular dimensions, e.g. irregularly distributed time steps?

2. lines 260-270: array databases by default store the data in a database, not in HDF5 or NetCDF, although some can be made to do so; they

may import data from HDF5 or NetCDF though. We believe the "Earth system scientists" mentioned in line 269 will soon be the old school if data cube access principles get more widely used beyond GEE and ESDC. No data scientist will want to go back to the individual files underlying a properly implemented data cube.

3. line 180: we think that spectral decomposition does not map from (time) to (time,freq), but to (freq). Eq (12) and (13) have the same problem.

4. Eq (15) para should be par?

5. line 500 ff: we believe that UDFs are quite widely spread and are implemented in SciDB, rasdaman commercial, openEO (GeoPySpark/GeoTrellis, Grass GIS), R package stars, and Python module xarray. Seeing an example of an ESDC UDF in the paper would be nice!

**References**

- Marius Appel, Edzer Pebesma, 2019, On-Demand Processing of Data Cubes from Satellite Image Collections with the gdalcubes Library. Data 4(3), 92

- Edzer Pebesma, Wolfgang Wagner, Pierre Soille, Miha Kadunc, Noel Gorelick, Matthias Schramm, Jan Verbesselt, Johannes Reiche, Matthias Mohr, Jeroen Dries, Alexander Jacob, Markus Neteler, Soeren Gebbert, Christian Briese and Pieter Kempeneers, 2019. openEO analyses Earth Observation data based on user-defined raster and vector data cube views. Geophysical Research Abstracts Vol. 21, EGU2019-9737, 2019, EGU General Assembly 2019. abstract, poster.

---

## Referee Comment (RC2) · Anonymous Referee #2 · 15 Nov 2019

**Review of "Earth system data cubes unravel global multivariate dynamics" by Mahecha et al. (esd-2019-62)**

November 15, 2019

In this article Mahecha et al. present the concept of data cubes to handle the growing body of Earth system data and introduce the computing interface "Earth system data lab" (ESDL) as a cloud-based solution. In the introduction the authors describe different data sources and variables of the Earth system, the hurdles of using the data, and illustrate the data cube approach as solution to the problem. Then, they delve into the concept and definition of data cubes, provide a generic description and mathematical formulations including how to apply customized operations on data cubes. In this context, Mahecha et al. explain the detailed implementation of the data cube approach in the ESDL project and depict its representation and processing of the various data streams. The authors showcase three example studies to demonstrate the functioning and usefulness of the ESDL.

The ESDL is novel and unique in its approach to focus on the fusion of global multivariate data streams and thus enables an simultaneous exploration of many facets of the Earth system. Therein, the ESDL is well equipped to face the challenges in the upcoming era of machine learning. Therefore, the ESDL and this descriptive article is an important contribution to the Earth system sciences and possibly to a wider community.

The manuscript is well structured and written. However, I have a few minor points of criticism that need consideration, before I can recommend this manuscript for publication.

**1 General Comments:**

1.1 *A large part of this article deals with the mathematical formulation and technical implementation of data cubes. However, you completely miss out on the technical description of the processing of the various data streams and how you treat uncertainty in the ESDL. For example, how do you treat the provided uncertainty estimates / quality flags of the individual data products? How does this effect the remapping / resampling algorithms? Is the ESDL capable to take error propagation into account? Could you provide a flow-chart to illustrate the procedure of how you incorporate data streams?*

1.2 *The second case study on the intrinsic dimension(s) of land surface variables clearly demonstrates the usefulness of the ESDL and thereby supports the title of the manuscript. Also the first case study corroborates the statement that multivariate dynamics can be better studied with data cubes, less convincingly though. However, I am not convinced that the third study really supports the need for the multivariate approach in ESDL. Here, you basically analyze only two variables (ecosystem respiration and temperature), which one could easily do with any other tool not based on data cubes. Can you make more clear why this case study supports the claim in the title?*

1.3 *The ESDL really lives on the various data streams. Many researchers have their specific datasets which they would want to analyze alongside the data streams provided in the ESDL environment. How do you enable the usage of external datasets? What are the disk usage constraints for each individual user? Is it possible to stream data (e.g. in Zarr format) from external data storage? If incorporating own datasets constitutes a complicated endeavor, researchers might be hesitant to use ESDL.*

**2    Specific comments:**

2.1 *L36: You cannot only analyze the state, but also the change of the system using ESDL, right?*

2.2 *L35: You start the paragraph claiming that we are well prepared in terms of data availability, but here you say there are access barriers. Maybe the term "availability" is not accurate here. A huge amount of data are collected, but they are not necessarily available for science. So, you could start this paragraph saying that we are well-prepared in terms of data collection.*

2.3 *L70: I suggest to remove 'we believe ...' and phrase the sentence: Due to its .... interface, the ESDL is well-suited ....*

2.4 *L86: This sentence ("However, ...") is somewhat complicated to grasp. I guess you want to motivate why "variable" should be treated as an additional dimension. Please revise this sentence.*

2.5 *L90: Here, the subscript of Y denotes the different variables k and the superscript denotes the different domains j - this is not consistent with the definition of Y in LL85-87.*

2.6 *L98: If the dimensions for only one grid point are dropped, do you not lose information? For example, a point measurement with certain lat and lon coordinates and a timestamp cannot be represented without losing the coordinate information, right?*

2.7 *L100: Here you use a math symbol / notation (large $\times$) to describe a cartesian product, which I have rarely seen before. Please mention that large $\times$ refers to cartesian product.*

2.8 *L105: Please define NA - does it refer to "not available", thus missing data?*

2.9 *L111: In the modelling community data cubes can be used to represent large ensembles, thus I suggest to also list "ensemble member" as another relevant dimension.*

2.10 *L214: If I may suggest to contact the maintainers of the Integrated Climate Data Center (ICDC, https://icdc.cen.uni-hamburg.de/daten.html) at the University of Hamburg. They do a very good job in collecting, processing (e.g. remapping, quality assurance), and maintaining/updating data of any kind relevant for Climate / Earth system sciences - provided in netCDF, which can easily be converted to Zarr. I assume one could join forces and build an even more comprehensive ESDL.*

2.11 *L219: "...given of their..." does not sound correct. I suggest to omit 'of' and write "... been ingested given their recurrent...".*

2.12 *LL237-247: This part explains some functions of the ESDL.jl toolbox. At this point, this information is not necessarily important for the reader. Maybe it is enough to refer to ESDL.jl documentation and the case studies which are accompanied with code - as you do it for the python implementation of ESDL.*

2.13 *L290: 'an univariate time series'*

2.14 *L291: I suggest the following corrections and revisions: "If one stored the same data cube with complete time series contained in one chunk, read operations could perform much faster."*

2.15 *L303: Delete first occurrence of 'behaviour'. I would also omit 'time', since it is redundant in the combination with 'long-term', i.e. 'long-term system behaviour in time'*

2.16 *L317: Please capitalize "northern" or use lowercase consistently for all occurrences of "northern" and "southern hemisphere".*

2.17 *L320: Please capitalize "fig" or use lowercase consistently for all occurrences.*

2.18 *L321: "Southern Hemisphere" in singular.*

2.19 *L348: I suggest using present tense when describing what you did in your study and using past tense when describing what others did before you, thus 'In our application, we follow this approach...'. Please use tenses consistently across the paper.*

2.20 *L352: Where do you introduce all the acronyms? Please provide the written-out terms here or refer to the table in the Appendix.*

2.21 *L353: I suggest to delete "the latter two", since it is not needed.*

2.22 *L361: I suggest to use the term "seasonal cycle" in singular.*

2.23 *L374: You use 'essentially', so delete 'only' in '... driven essentially by solar forcing only...".*

2.24 *L377: I recommend to use 'complex' instead of 'complicated'.*

2.25 *LL377-379: This sentence ('Zooming...') is somehow complicated to read and grasp. Maybe you can split the sentence and provide more information why it is important/interesting to focus on the northern regions of South America.*

2.26 *L382: Please use 'land surface' or 'land-surface' consistently throughout the paper.*

2.27 *L392: Here you put $R_\mathrm{eco}$ in italic letters and earlier (e.g. L352) you don't. Please use math notation consistently and follow the conventions explained in the Copernicus LaTeX template.*

2.28 *L402: Eq. (15) is incomplete. The minimal output dimensions are 'para' and 'time'. Where is the time term in the equation? Also, correct typo 'par' to 'para'.*

2.29 *L405: Please check for consistent usage of 'high-latitude, high-dimensional, high-resolution, etc.', so, with or without hyphen.*

2.30 *L418: Correct typo 'supporting materials'.*

2.31 *L426: Please check for consistent usage of 'semi-arid' versus 'semiarid' (L428).*

*2.32 L461: Where are these simplifications described in detail?*

*2.33 L464: Can 'would be' replaced by 'is'?*

*2.34 LL465-469: Please consider my comment again w.r.t. to contacting the ICDC maintainers (Comment 2.10)!*

*2.35 L510: Delete one 'several'.*

*2.36 L512: Maybe better '..., with no claim to completeness, ..." or '..., without claiming completeness, ...'.*

*2.37 L525: Please explain shortly 'kaggle' or provide an URL / reference.*

*2.38 L530: Delete one 'the'.*

*2.39 L535: The straightforward implementation of ESDL to handle / analyze CMIP multi-model but also the emerging grand ensembles of several hundreds of simulations is a key strength in the ESDL approach, in my opinion. I suggest to promote this aspect more strongly and include respective keywords, such as 'multi-model ensemble', 'large' or 'grand ensemble' in the abstract.*

*2.40 L557: Delete 'on'.*

*2.41 L557: "Dimension" in plural.*

*2.42 L594: The last sentence of the conclusion section is somewhat cumbersome. Can you boil down or split this sentence? As a reader, I expect the last sentence of the paper to be a strong and precise statement.*

*2.43 Figure 2: This is certainly an appealing visualization, however, it does not convey much information. Maybe this is also the reason why do not reference this figure anywhere in the text. I suggest to reconsider if this figure is really needed or if the url to the animation or providing the animation as ESD asset accompanying the article is sufficient. If the figure is needed, than I suggest to make some modifications as illustrated in Figure 1 and add a legend so that the figure is understandable without studying the details in the caption.*

[Figure]

Figure 1: Modifications for Figure 2

2.44 *Figure 3: The actual values represented by the gray color-scale are not visible at all and are thus redundant in the current visualization. I suggest to include labels for actual values in the polar plot, as described here: [https://matplotlib.org/gallery/pie_and_polar_charts/polar_legend.html#sphx-glr-gallery-pie-and-po](https://matplotlib.org/gallery/pie_and_polar_charts/polar_legend.html#sphx-glr-gallery-pie-and-po)*

2.45 *Figure 4: Please explain why there are no data for the Arctic.*

2.46 *Figure 5: Better vertically center the y-label.*

2.47 *Figure 6: Please correct typos: "the latter reduces …. and leads …"*

---

## Author Comment (AC1) · 18 Dec 2019

**The comments of the reviewer are repeated here in bold font;** *our answers are given in italics.*

**The authors present a new data cube approach, called Earth System Data Cube, where multiple spatiotemporal data streams are treated as one singular, very high-dimensional data stream. This paper is of high quality and clearly outlines the authors' reasoning of implementing the ESDC in the way it is implemented and the advantages it brings to Earth System dynamics studies. The ESDC datacube approach brings different data streams to a common grid, which is beneficial for specific scientific Earth System studies. Common operations, as e.g.**

[Figure]

**resampling and bringing the data to the same spatial and temporal resolution, are just taken care of and users can focus more on science. Yet, the unified grid from the beginning might be a limitation for other application areas.** *We thank the reviewer for the very positive comments and for sharing her/his concerns w.r.t. the technical approach chosen here. Re-gridding the data to a common format was essentially necessary to start thinking in "cubes" where common axes are indeed needed. Otherwise the formalism, but more importantly, the implementation would have been substantially more complicated, yet not impossible. But we agree that this can be regarded as suboptimal for certain applications. We will, in the revised version, discuss the potential drawback of a pre gridded data set and also differentiate better between the "concept" of the ESDL and the "implementation". In fact, the latter can be extended to deal with data of different grids and then work on the mismatch on the fly.*

**Some comments and areas for minor revision:**

**What is not so clear for me so far is the overall implementation strategy of the ESDC. I understand that I can now use the ESDC with the datasets as outlined in the paper appendix. Is there the option that users can extend the ESDC data list?** *Indeed, as described by in the paper there is the possibility to 1) use the ESDL with the current data, 2) add more variables to an existing cube, or 3) use the code implementation with own data.*

**Or will it be similar to the OpenDataCube concept where multiple implementations of the ESDC concept can be set up with different data?** *As we describe we have already several cubes (see L211 and L215) implemented and more can follow (L298). See also our discussion on future perspectives section 5.3 L540.*

**Where is the current ESDC data hosted?** *We quote from our paper, page 13; table 2: "The cubes are currently hosted on the Object Storage Service by the Open Telecom Cloud under https://obs.eu-de.otc.t-systems.com/obs-esdc-v2.0.0/ "*

**And how long does the data preparation take?** *There is no generic answer to this*

*question as it depends on the native format of the data that should be ingested. If the data are already in suitable NetCDF, it is essentially a very fast conversation to the currently supported zarr-Format. It also depends on the size and other aspects.*

**I suggest to bring in these additional aspects.** *We thank you for the suggestion and will focus on making the points on "own data" more clear. Given that we have had alreday addressed the aspects, we simply will try to describing them more prominently in the text.*

**What are the bottlenecks in gathering the data? I understand that in order to implement the ESDC, data has to be moved from the respective data repositories. Is this correct? Would be good to elaborate on this aspect as well.** *Yes, we need a common repository - or at least a common format that serves the data in a joint data model as said in L64. For details on the storage we had written Section 3.3 (L 260).*

**It is without doubt that Julia is a powerful and efficient programming language. However, the fact that the ESDC package has been developed in Julia could be a restrictive factor in the uptake, as I argue that most potential users of the ESDC use Python or R and might be less motivated to take up a new programming language. Are there plans to extend it to Python or R?** *The reviewer is right that many users would prefer Python and R over Julia. But please note that in fact the ESDL is perfectly working from python. As we write in L. 258 and 279, we have a running Python interface that can be inspected here https://cablab.readthedocs.io/en/latest/. Working with R is more complicated as there is no suitable implementation of the zarr-format available yet. In response to your comment and other requests, one of the co-authors of this paper (Guido Kraemer) is working on an implementation for reading the zarr format in R (https://github.com/gdkrmr/zarr-R), this can serve as a basis for a future implementation of the data cube framework in R. In particular we aim to use data cube concepts as developed e.g. by Edzer Pebesma and Marius Appel (see commentary uploaded by them to this discussion) in the stars framework (https://github.com/r-spatial/stars) to talk with our data-cubes. In response to the comment we will mention the Python*

*implementation more prominently in the revised version and also point the users to the R developments in progress.*

**The authors argue that the ESDL is closest to the Climate Data Store. In my current understanding of both data systems, I would disagree. Do the authors talk about the Climate Data Store, which is primarily a data dissemination system, or the Climate Data Store toolbox, which is the processing editor on top of the CDS? It would be necessary to elaborate more on what aspects both systems are close and why the authors come to this assessment. It is also important that the authors differentiate between the CDS and the CDS toolbox.** *Thank you for the advice to be more precise on this topic. We will elaborate this in the revision. Our point is that the CDS indeed does offer also data analytic access via jupyter notebooks that allow the user to map UDFs as we do on an arbitrary set of of data stored there.*

**Some additional typing errors discovered: ...** *Thank you for spotting these errors - we will work on these in the revisions.*

---

## Author Comment (AC2) · 19 Dec 2019

**Comments of the reviewer are pasted here in bold font;** *our answers are given in italics.*

**In this article Mahecha et al. present the concept of data cubes to handle the growing body of Earth system data and introduce the computing interface "Earth system data lab" (ESDL) as a cloud-based solution. In the introduction the authors describe different data sources and variables of the Earth system, the hurdles of using the data, and illustrate the data cube approach as solution to the problem. Then, they delve into the concept and definition of data cubes, provide a generic description and mathematical formulations including how to apply**

**customized operations on data cubes. In this context, Mahecha et al. explain the detailed implementation of the data cube approach in the ESDL project and depict its representation and processing of the various data streams. The authors showcase three example studies to demonstrate the functioning and usefulness of the ESDL. The ESDL is novel and unique in its approach to focus on the fusion of global multivariate data streams and thus enables an simultaneous exploration of many facets of the Earth system. Therein, the ESDL is well equipped to face the challenges in the upcoming era of machine learning. Therefore, the ESDL and this descriptive article is an important contribution to the Earth system sciences and possibly to a wider community. The manuscript is well structured and written. However, I have a few minor points of criticism that need consideration, before I can recommend this manuscript for publication.** *We thank the reviewer for the precise summary of the ESDL and enthusiasm with respect to the potential of this approach. We will address all points in the remainder of the review in due detail.*

**1 General Comments:**

**1.1 A large part of this article deals with the mathematical formulation and technical implementation of data cubes. However, you completely miss out on the technical description of the processing of the various data streams and how you treat uncertainty in the ESDL. For example, how do you treat the provided uncertainty estimates / quality flags of the individual data products? How does this affect the remapping / resampling algorithms? Is the ESDL capable to take error propagation into account? Could you provide a flow-chart to illustrate the procedure of how you incorporate data streams?** *The reviewer is right that this paper has been written with an emphasis on the conceptual aspects of the "data cube idea". Many concepts for data cube have been proposed in the last few years in the geo community and many are still under active development. However, we feel that we are still lacking an overarching vision of how data cubes can empower Earth system sciences.*

*Regarding the specific question on the uncertainty: We have to admit that we have had many discussions on how to consider uncertainty and propagate it to the ESDL. We found, however, that each data product comes with its own uncertainty, some e.g. with flags, others with confidence bands, and other entirely without information on the uncertainty. Hence, we couldn't come up with a unified view on an "uncertainty dimension" in the ESDL framework that could give due credit to all of them. The procedure to incorporate a data stream is extensively described in the documentation of the ESLD:* `https://cablab.readthedocs.io/en/latest/esdc_prod.html` *- which will be mentioned in the revisions of the paper. Regarding the last point on adding a flow-chart we agree that his is a good idea and will add it in a revision version of the paper.*

**1.2 The second case study on the intrinsic dimension(s) of land surface variables clearly demonstrates the usefulness of the ESDL and thereby supports the title of the manuscript. Also the first case study corroborates the statement that multivariate dynamics can be better studied with data cubes, less convincingly though. However, I am not convinced that the third study really supports the need for the multivariate approach in ESDL. Here, you basically analyze only two variables (ecosystem respiration and temperature), which one could easily do with any other tool not based on data cubes. Can you make more clear why this case study supports the claim in the title?** *Thank you for this important comment. We understand that addressing a two-variable problem is not that convincing. The rationale for this use-case is actually to show that there is no limit to address problems beyond "data exploration" in the ESDL. To put it in other terms: this example was also chosen for its simplicity (although it is not trivial), but can serve as an example for implementing a parameterizing more complex models. We will clarify and discuss this in our revision.*

**1.3 The ESDL really lives on the various data streams. Many researchers have their specific datasets which they would want to analyze alongside the data streams provided in the ESDL environment. How do you enable the usage of ex-**

ternal datasets? **What are the disk usage constraints for each individual user?**
**Is it possible to stream data (e.g. in Zarr format) from external data storage?**
**If incorporating own datasets constitutes a complicated endeavor, researchers**
**might be hesitant to use ESDL.** *This concern is one of the most often raised ques-*
*tions in the various user consultation meetings we had so far and was also raised by*
*reviewer 1. To give a brief answer: Yes, it is indeed possible to add "own" data sets any*
*pre-curated data cube. One can, for instance, read any xarray data set - as long as it*
*shares common axis with the existing cube. In the experimental platform that was set*
*up now we have, of course, disk usage constraints, but the paper describes a generic*
*system that is independent of the concrete jupyter hub running right now. We also*
*note that the implementation in Julia and the Python based xarray allow for reading in*
*additional NetCDF files and concatenating them with an existing cube (again - under*
*the assumption of shared axes). We can also read additional cubes into memory.*

**2 Specific comments:**

**2.1 L36: You cannot only analyze the state, but also the change of the system**
**using ESDL, right?** *Yes indeed! For an example study in this direction we refer to a*
*paper in discussion by our co-author Guido Kraemer et al. https://www.biogeosciences-*
*discuss.net/bg-2019-307/bg-2019-307.pdf i.e. figures 5 and 6 therein. But we will*
*mention this now also in this manuscript more prominently.*

**2.2 L35: You start the paragraph claiming that we are well prepared in terms**
**of data availability, but here you say there are access barriers. Maybe the term**
**"availability" is not accurate here. A huge amount of data are collected, but**
**they are not necessarily available for science. So, you could start this paragraph**
**saying that we are well-prepared in terms of data collection.** *Thank you for this*
*sharp observation! We will change this accordingly.*

**2.3 L70: I suggest to remove 'we believe ...' and phrase the sentence: Due to**
**its .... interface, the ESDL is well-suited ....** *We agree and will change the text*

*accordingly.*

**2.4 L86: This sentence ("However, ...") is somewhat complicated to grasp. I guess you want to motivate why "variable" should be treated as an additional dimension. Please revise this sentence.** *We agree and will change the text accordingly.*

**2.5 L90: Here, the subscript of Y denotes the different variables k and the superscript denotes the different domains j - this is not consistent with the definition of Y in LL85-87.** *Thank you for spotting this! The first author apologizes for sloppiness to his co-authors as this remark has been stated internally before. We will change the text accordingly.*

**2.6 L98: If the dimensions for only one grid point are dropped, do you not lose information? For example, a point measurement with certain lat and lon coordinates and a timestamp cannot be represented without losing the coordinate information, right?** *Well in fact it can. We have chosen this notation for the sake of simplicity and because this is how most programs of scientific computing deal with such phenomena. But one can always define the mapping such that the collapsed dimension retains a length of 1. The risk we see is that very long workflows become intractable as there is no "simplification" in the dimensionality. But it is a bit irritating at times We will add a remark in the text to clarify that both ways are thinkbale but that we prefer this approach here.*

**2.7 L100: Here you use a math symbol / notation (large $\times$) to describe a cartesian product, which I have rarely seen before. Please mention that large $\times$ refers to cartesian product.** *We will add a footnote clarifying the meaning of this symbol.*

**2.8 L105: Please define NA - does it refer to "not available", thus missing data?** *Yes, will clarify this in the text accordingly. See also response to comment by Apel and Pebesma.*

**2.9 L111: In the modelling community data cubes can be used to represent large ensembles, thus I suggest to also list "ensemble member" as another relevant dimension.** *Please note that we have this already in the paper cf. lines 532 and 539. But we agree that it should be listed in L111 as well.*

**2.10 L214: If I may suggest to contact the maintainers of the Integrated Climate Data Center (ICDC, `https://icdc.cen.uni-hamburg.de/daten.html`) at the University of Hamburg. They do a very good job in collecting, processing (e.g. remapping, quality assurance), and maintaining/updating data of any kind relevant for Climate / Earth system sciences - provided in netCDF, which can easily be converted to Zarr. I assume one could join forces and build an even more comprehensive ESDL.** *Thank you for the hint. Indeed we have had various points of contact with Hamburg and very much hope that future collaborations can emerge in the direction you suggest here.*

**2.11 L219: ' ... given of their ... " does not sound correct. I suggest to omit 'of' and write "... been ingested given their recurrent ... ".** *We agree and will change the text accordingly.*

**2.12 LL237-247: This part explains some functions of the ESDL.jl toolbox. At this point, this information is not necessarily important for the reader. Maybe it is enough to refer to ESDL.jl documentation and the case studies which are accompanied with code - as you do it for the python implementation of ESDL.** *We note that this request to remove this part stands in opposition to the comment posted by Prof. Pebesma and Dr. Appel who actually requested us to extend this part. We think this is an editorial decision and look forward to the opinion of the handling editor to proceed accordingly. We agree, however, that software developments can change rapidly so that it could be better having their description separated from the scientific concepts.*

**2.13 L290: 'an univariate time series'** *We agree and will change the text accordingly.*

**2.14 L291: I suggest the following corrections and revisions: "If one stored the same data cube with complete time series contained in one chunk, read operations could perform much faster."** *We agree and will change the text accordingly.*

**2.15 L303: Delete first occurrence of 'behaviour'. I would also omit 'time', since it is redundant in the combination with 'long-term', i.e. 'long-term system behaviour in time'** *We agree and will change the text accordingly.*

**2.16 L317: Please capitalize "northern" or use lowercase consistently for all occurrences of "northern" and "southern hemisphere".** *We agree and will change the text accordingly.*

**2.17 L320: Please capitalize "Fig" or use lowercase consistently for all occurrences.** *We agree and will change the text accordingly.*

**2.18 L321: "Southern Hemisphere" in singular.** *We agree and will change the text accordingly.*

**2.19 L348: I suggest using present tense when describing what you did in your study and using past tense when describing what others did before you, thus 'In our application, we follow this approach ...'. Please use tenses consistently across the paper.** *We agree and will change the text accordingly.*

**2.20 L352: Where do you introduce all the acronyms? Please provide the written-out terms here or refer to the table in the Appendix.** *All acronyms were introduced in section 3.1. I.e. way before L352 and some again at the beginning of Section 4.*

**2.21 L353: I suggest to delete ' the latter two", since it is not needed.** *We agree and will change the text accordingly.*

**2.22 L361: I suggest to use the term "seasonal cycle" in singular. 2.23** *We agree and will change the text accordingly.*

**L374: You use 'essentially', so delete 'only' in "... driven essentially by solar**

**forcing only ...".** *We agree and will change the text accordingly.*

2.24 L377: I recommend to use 'complex' instead of 'complicated'. *We respectfully disagree as the notion of complexity is not trivial and would require additional analytics that are beyond the scope of this paper.*

**2.25 LL377-379: This sentence ('Zooming...') is somehow complicated to read and grasp. Maybe you can split the sentence and provide more information why it is important/interesting to focus on the northern regions of South America.** *We agree and will change the text accordingly, i.e. we will provide a rationale for our interest in this region.*

**2.26 L382: Please use 'land surface' or 'land-surface' consistently throughout the paper.** *We agree and will change the text accordingly.*

**2.27 L392: Here you put Reco in italic letters and earlier (e.g. L352) you don't. Please use math notation consistently and follow the conventions explained in the Copernicus LaTeX template.** *We actually were actually unsure about this, as we have not written variable names in italic, but here need it as mathematical symbol. We will go to the Copernicus style guide and change the text accordingly.*

**2.28 L402: Eq. (15) is incomplete. The minimal output dimensions are 'para' and 'time'. Where is the time term in the equation? Also, correct typo 'par' to 'para'.** *Thank you for spotting this! This is inherited from an earlier version of the paper where we had a more complex approach. We will change the text accordingly.*

**2.29 L405: Please check for consistent usage of 'high-latitude, high-dimensional, high-resolution, etc.', so, with or without hyphen.** *We agree and will change the text accordingly.*

**2.30 L418: Correct typo 'supporting materials'.** *We agree and will change the text accordingly.*

**2.31 L426: Please check for consistent usage of 'semi-arid' versus 'semiarid'**

**(L428).** *We agree and will change the text accordingly.*

**2.32 L461: Where are these simplifications described in detail?** *We agree and will change the text accordingly.*

**2.33 L464: Can 'would be' replaced by 'is'?** *We prefer to keep this wording as is as the statement is of a speculative nature.*

**2.34 LL465-469: Please consider my comment again w.r.t. to contacting the ICDC maintainers (Comment 2.10)!** *Yes, indeed!*

**2.35 L510: Delete one 'several'.** *We agree and will change the text accordingly.*

**2.36 L512: Maybe better '... , with no claim to completeness, ...' or '..., without claiming completeness, ...'.** *We agree and will change the text accordingly.*

**2.37 L525: Please explain shortly 'kaggle' or provide an URL / reference.** *We agree and will change the text accordingly.*

**2.38 L530: Delete one 'the' .** *We agree and will change the text accordingly.*

**2.39 L535: The straightforward implementation of ESDL to handle / analyze CMIP multi-model but also the emerging grand ensembles of several hundreds of simulations is a key strength in the ESDL approach, in my opinion. I suggest to promote this aspect more strongly and include respective keywords, such as 'multimodel ensemble', 'large' or 'grand ensemble' in the abstract.** *We cannot agree more and will promote it in the abstract accordingly. Please note that in fact, co-author Fabian Gans has implemented a prototype that can be explored online as shown in this gist: https://gist.github.com/meggart/2d544be2c1368f8774d0a21ea4633985. However, this is still work in progress in collaboration with the Pangeo community so we did not include it in this paper.*

**2.40 L557: Delete 'on'.** *We agree and will change the text accordingly.*

**2.41 L557: "Dimension" in plural.** *We agree and will change the text accordingly.*

**2.42 L594: The last sentence of the conclusion section is somewhat cumbersome. Can you boil down or split this sentence? As a reader, I expect the last sentence of the paper to be a strong and precise statement.** *We agree and will have to think about it and will change the text accordingly.*

**2.43 Figure 2: This is certainly an appealing visualization, however, it does not convey much information. Maybe this is also the reason why do not reference this figure anywhere in the text. I suggest to reconsider if this figure is really needed or if the url to the animation or providing the animation as ESD asset accompanying the article is sufficient. If the figure is needed, than I suggest to make some modifications as illustrated in Figure 1 and add a legend so that the figure is understandable without studying the details in the caption.** *We admit that it was a mistake not to reference the figure properly in the text. However, we respectfully disagree with the statement that this figures does not convey information and hence would like to ask the editor for permission to keep this figure without further modification. The rationale is the following: This figure is a new variant of an earlier figure published here https://figshare.com/articles/Earth_Data_Cube/4822930 which has been used widely. For instance we found many copies of it at the last EGU conference posters (actually without proper citation). Hence, we believe that it is a very suitable means to convey the fundamental idea of the paper in a conceptual manner. If we would go for a technical illustration as you suggest it here with axes labels or colorbars, units etc. it would lose the character of a conceptual figure Hence, we would like to suggest to keep it and explain its purpose more accurately + referencing it properly in the text.*

**2.44 Figure 3: The actual values represented by the gray color-scale are not visible at all and are thus redundant in the current visualization. I suggest to include labels for actual values in the polar plot, as described here: https://matplotlib.org/gallery/pie_and_polar_charts/polar legend.htmlsphx-glr-gallery-pie-and-polar-charts-polar-legend-py** *We had this*

*discussion as well among coauthors. We had a version with labels as you suggested, but these were not readable in print either. Hence, we prefer the current version as the colorbar gives exactly the range and allows us to annotate the usints properly.*

**2.45 Figure 4: Please explain why there are no data for the Arctic.** *We will explain this with data fractionations.*

**2.46 Figure 5: Better vertically center the y-label.** *We believe that the esthetics is up to the authors.*

**2.47 Figure 6: Please correct typos: ' the latter reduces ... and leads..."** *We agree and will change the text accordingly.*

*As a final sentence we would like to thank the reviewer again for the very detailed feedback that will greatly improve the quality of the manuscript! We will acknowledge this is the revised paper as well.*

---

## Author Comment (AC3) · 20 Dec 2019

**Comments by Appel and Pebesma are pasted here in bold font;** *our answers are given in italics.*

**We congratulate the authors of this paper with a very nice contribution describing the ESDC and its application, an activity that has involved a substantial conceptual development as well as implementation work, that has triggered a number of scientific papers already, and that now helps furthering the discussion what spatiotemporal data cubes are. We are mostly interested in having this latter discussion.** *We are very pleased about this contribution to the discussion from two leading experts in the field of data cubes. The points raised here will carefully be*

[Figure]

*considered in the revision of the manuscript wherever possible.*

**1 Pre-grid all the data, or do this on-the-fly?**

**The ESDC pre-grids all the data, and (l456) "One of the most commonly expressed practical concerns is the choice of a unique data grid". Given that an ESDC is defined on a single grid, but integrates a lot of different datasets, each of these datasets have to be forged into the target grid. This involves resampling, statistical downscaling, interpolation and/or aggregation of data both in space and time, as well as handling coordinate reference systems and possibly different calendars (Gregorian, 365day/noleap, 360 day) into one. It is unclear whether the ESDC can do all this, and also if it can give an idea about the errors introduced by doing so, or whether it can give users otherwise recommendations what a good grid is? Other systems, e.g. Google Earth Engine or Sentinel Hub work from the raw data (which can be in many different coordinate reference systems, e.g. Sentinel-2 distributed over 120 UTM zones), and create user-defined data cubes on the fly. This has the advantage that low-resolution computations can be done relatively fast and interactively, which helps data exploration and model development, and that the effect of different target resolutions can be easily evaluated. With the ESDC, once data are cubed, users no longer have the possibility to go back to the original data. We think this issue is important, and missed a discussion on this matter in the paper. Given that large, operational systems exist that do not store precubed datasets but that work with data cube views (Pebesma et al., 2019) we believe that this may be at least a viable option.** *At first glance, the concern raised here seems to be of rather fundamental nature. But after some discussion we think it is probably not that critical for most aspects of paper. First of all, the question of regridding is not at all questioning the fundamental principles as outlined in section 2 (the concept we promote here) and rather concerns the concrete implementation described in section 3.2. With respect to the implementation we would like to emphasize that we too see a lot of potential strategies that need to*

*be explored. There is clearly much room for improvement. Some thoughts on the specific question of pre-gridding versus on the fly pre-processing: Systems that allow for on-the-fly regridding would probably need to cache access to data that come in a resolution that is lower than requested. Otherwise, such a system would have to re-read all the data for every computation. We therefore like to think that the pre-curation of a cube in different resolutions can be conceptually seen as an explicit "cache". A cache that has been created with some fundamental considerations on how it is preprocess (e.g. guaranteeing mass balances when later summing up). Yet another point is that we don't really understand how fast computations can be achieved when aggregating on the fly. But in essence we admit that we have also a historical legacy: When we started implementing the first versions of the ESDL a few years back, we had the choice to either to regrid the data and quickly come to an usable cube, or to invest much more time in the computational developments. Our practical decision was to go for the regridding, and focussing all our energy on the conceptual multidimensional aspects of the implementation as requested by the scientific community. In fact we had several user consultation meetings along these lines. Still, and for us this is important to communicate here, we are not at all against considering different solutions to the problem and considering user-defined on-the-fly regridding. In fact, we had already several discussions among co-authors regarding this aspect. What we are missing so far for such an approach is a suitable storage backend library that can deal with this and would allow us to offer this approach.*

**2 Is latitude the same as time?**

**In Line 598, the paper states that "The ESDL is probably the most radical data cubing approach", and refers to some grey literature that also claims that data cubes should treat all dimensions identically, irrespective their semantics. We believe however that space and time are inherently different, and need different treatment. The example (e.g. fig 3) shows that all longitudes are aggregated to give profiles per latitude and time, but we feel that this is a rather contrived ex-**

**ample; in general, any transect in space could be a good candidate for reducing space to one dimension, and this would require mapping onto that transect; it is not so likely that you would want to do that on an arbitrary transect in the two-dimensional space defined by e.g. (longitude, time). In general, a large number of functions applied to space operate on both spatial dimensions (e.g., polygonal crop), and time series models are of a very different kind and are rarely applied to single spatial dimensions. The fact that you can do this is nice, but we do not believe it more of a big selling point than an opportunity for users to shoot themselves in the foot.** *Thank you for this important comment which has many aspects to discuss. Let's start with the least critical point: We agree that one should not really refer to gray literature, but in this very case the Data Cube manifesto did come out in parallel to our developments and we thought that it would be appropriate to cite outcomes from projects that have thought along similar lines, even if their implementation philosophy is very different than ours. The question whether space and time have to be regarded as inherently different, however, is not that obvious to us. We agree that one cannot compare these aspects in physical and philosophical terms, but, and this is important for us, to the computer they can look identical. This makes our lives easier, as we can write any UDF without thinking about the characteristic data-properties of the different dimensions. We also don't agree with the perception that Fig. 3 is a contrived example. There are many examples where such statistics are used, e.g. when averaging primary production by latitude and then repeating this for different periods of the year. Another example are the famous "flying carpets" (e.g. $CO_2$ concentrations as a function of time and latitude) e.g. at* `http://www.esa-ghg-cci.org/?q=node/115`*. In other terms, many standard applications in the Earth system sciences do make it necessary to have equal access to space and time. Another prominent example is certainly the Hovmöller diagram type. And yes, why not applying a PCA on this to retain the underlying orthogonal components only? Dimensionality reduction applications can well require either considering space and time or time and variable or space and variables in a single framework. Furthermore, we would argue that many relevant operations*

*can be reduced to convolutions (in space or time) and formulated for the 1D, 2D or 3D case. It is very important for us to emphasize these points, because we want to firmly reject the argument that we are looking for ' selling points". Of course there are models that only make sense in a temporal context, e.g. causal time series models. But we trust that any scientist applying a model of this kind is sufficiently knowledgeable to also understand what dimensions can be addressed with it.*

**3 Is a lat/long grid the only way we can cube the Earth?**

**No, it isn't, and many datasets use other global grids, discrete global grids, or collections of grids (Equi7, or UTM zones). All this has reasons, and the Earth will never be flat so the problem will remain. A discussion on generalizing the ESDC to other grids, or collections of these, would be welcome.** *Indeed it isn't. We have a bias of thinking in lat-long-worlds because we are coming from a branch of science where basically all widely used datasets are distributed on a such a longitude-latitude grid. But we totally agree that other grids are likewise highly relevant. We will discuss this accordingly in the revised manuscript.*

**4 Vector data cubes are missing**

**In the representation of the ESDC, the implicit assumption is being made that data cubes correspond to spatial raster data. This is not the case: space can be a one-dimensional set of feature geometries (e.g. points, or polygons). One of the most requested feature of data cubes is to retrieve all the cube information at a given set of points, or aggregated over a given set of polygons. This leads naturally to vector data cubes. Can ESDC answer such queries, or do the authors consider this to not be data cubes?** *This question touches a common request. In order to show that we can likewise deal with other vector data cubes we are elaborating a fourth use case that we will include either in the revised version of the paper or the appendix. In short, our answer we consider these data cubes and we can deal with them.*

**5 Where is support?**

**Spatial grids may refer to a collection of points on a regular grid, or to a collection of grid cells as if they were small square polygons. In the latter case, properties can be either continuous over a grid cell (e.g., land use, soil type, geology) or may be aggregated values over the grid cell (e.g., the total amount of carbon in the grid cell, or its maximum elevation). Similarly can time be conceived as a set of time instances or intervals, with the two interval interpretations. Does the ESDC take care of some of these options, or is this all assumed to be remembered by the users?** *In the current form, we do not treat such grid cells differently. There are, however, developments that will try to get this solved in the near future cf. https://github.com/JuliaGeo/DimensionalArrayTraits.jl.*

**6 Code and reproducibility**

**We are happy to learn that the software is open source, and can be used by others. Have the authors heard success stories of others installing and using the software? A few small code examples in the paper to get a taste for how simple queries to the ESDC look like would have worked well, and could be encourage those hungry for trying it out. We did not try out the Julia script but hope that other reviewers will report whether they did, and whether they were successful in reproducing the results shown in the paper.** *Yes, we have finalized an experiment within an "Early Adopter Call" funded by ESA where students from the B.A. to the PhD level had a few weeks of time to work with the ESDL system. The results showed a very broad way to utilize it. For instance, one project adapted the ESDL to ingest high resolution Sentinel 1 data. This shows us that it seems feasible to work with the software. Regarding the question for "small code examples" we have in fact the opposite request from reviewer 2 - removing the implementation part. So we think the current solution of having real code examples on a git is better as it also has more flexibility compared to having them in a paper.*
**7 Dropping dimensions**

**If a dimension has only one value, it is dropped; we can see this is useful, but it also drops the information of the dimension (e.g. the species name, or the elevation value). We see a lot of 4-dimensional NetCDF files in the wild having 1 dimension with one values, to specify the value for that dimension, which seems all but useless. You need to be able to drop it, but can the ESDC also not drop it, e.g. so that sub-cubes can be meaningfully combined along the otherwise dropped dimension?** *A comparable comment has been raised by reviewer 2. To reiterate our answer: We have decided for dropping dimensions "as default" for keeping data cubes that result from some operation at their minimal dimensionality. We are aware that this approach also means losing information. For the example you give: you have e.g. a cube of lat, lon, species presence. If you would subset the cube and only retain the presence-absence of one species in a single cube we would actually expect the user to name the cube accordingly. That aside: the main point we have to make is that there is no problem in outputting a dimension of length 1. This can be easily integrated in both our notation and implementation if desired. For the latter, the* `mapCube` *function can do exactly this if the output dimension would be given with length 1.*

**8 The data model**

**Having a data model that maps from dimensions to IR and NA is great; a similar approach was adopted in Appel and Pebesma (2019). For end users it also means there is no way to properly handle logical (TRUE/FALSE or NA), categorical, or e.g. time variables. Of course 8-byte doubles can encode anything, but everyone who has tried to use them for encoding categories knows the nightmare. Do end users need that? Some sort of a discussion on this issue would be welcome.** *Any number of logical type can be represented here and we just tried to propose a notation that is "close" to what we suspect is the expectation of the majority of potential users.*

**9 Other issues**

**1. Can the ESDC cope with irregular dimensions, e.g. irregularly distributed time steps?** *In principle, i.e. if the irregularity is consistent across all other dimensions, yes. We simply need a dimension e.g. an irregular time step that repeats across variables or model runs or whatever property the other dimensions encode. If we would have different time-steps in different variable we have the same issue as discussed above on dealing with different lat-long grids and we are back to the question if we should enable on-the-fly regridding.*

**2. lines 260-270: array databases by default store the data in a database, not in HDF5 or NetCDF, although some can be made to do so; theymay import data from HDF5 or NetCDF though. We believe the "Earth system scientists" mentioned in line 269 will soon be the old school if data cube access principles get more widely used beyond GEE and ESDC. No data scientist will want to go back to the individual files underlying a properly implemented data cube.** *Well, but it is important for us that one can easily understand where the data are. . . But yes, people may not do that if there would be "properly implemented data cubes" indeed.*

**3. line 180: we think that spectral decomposition does not map from (time) to (time,freq), but to (freq). Eq (12) and (13) have the same problem.** *You would be right, if we had analyzed the power spectrum. But, as we write in the text this is about "discrete" decompositions via FFT. We will elaborate the text better to make this crystal clear.*

**4. Eq (15) para should be par?** *We used `para` in all points of the paper where we talked about parameters. `par` would be interpreted by many people as "Photosynthetically Active Radiation".*

**5. line 500 ff: we believe that UDFs are quite widely spread and are implemented in SciDB, rasdaman commercial, openEO (GeoPySpark/GeoTrellis, Grass GIS), R package stars, and Python module xarray. Seeing an example of an ESDC**

**UDF in the paper would be nice!** *All of these examples allow definition of UDFs, but they are in our mind not yet first-class citizens of the ecosystem. For us a UDF should be applicable on every combination of input axes and be equally efficient as applying "built-in" functions. For example, when applying a user-defined function in* `xarray+dask (xarray.apply_ufunc)` *with time as a core dimension. In this case the user is still limited to functions that are efficiently operating on many spatial data points at once, i.e. which can be expressed using vectorized numpy operations. Otherwise, if the user wants to apply a function which is defined to work on single time series only over a spatial grid, then this is possible (setting the argument vectorize=True), but will have significant performance overhead. Here we think that our Julia implementation is quite unique, because the user can formulate the algorithm in the most natural way (writing a function that operates on a single time series) and can apply this to cubes of any dimensionality with negligible overhead. One of the best examples is the definition of the sufficient_dimensions function, which can be found here:* `https://bit.ly/2ZgUltT`. *Here we apply an intermediately complex method on multivariate time series for each spatial location on a data cube which does not fit into the computer's memory. The whole operation is defined and applied with only a few lines of code using multiple processes and runs reasonably fast.*

*Finally, we would like to thank Dr. Marius Appel and Prof. Edzer Pebesma again for their contribution to this discussion. We will acknowledge their great comments also in the revised version of the paper.*

---

## Author Response (AR1)

**Earth system data cubes unravel global multivariate dynamics**

[revised manuscript text omitted]

As long as we do not overcome data interoperability limitations, Earth system sciences cannot fully exploit the promises of novel data-driven exploration and modelling approaches to answer key questions related to rapid changes in the Earth system (Karpatne et al., 2018; Bergen et al., 2019; Camps-Valls et al., 2019; Reichstein et al., 2019). A variety of approaches has been developed to interpret Earth observations and  big data in the Earth system sciences in general (for an overview see e.g. Sudmanns et al., 2019), and gridded spatiotemporal data as a special case (Nativi et al., 2017; Lu et al., 2018). For the latter, data cubes have become recently popular addressing an increasing demand for efficient access, analysis, and processing capabilities for high-resolution remote sensing products. The existing data cube initiatives and concepts  (e.g. Baumann et al., 2016; Lewis et al., 201 in their motivations and functionalities. Most of the data cube initiatives are, however, motivated by the need for accessing singular (very) high resolution data cubes, e.g. from satellite remote sensing or climate reanalysis, and not by the need for global multivariate data exploitation.

This paper has two objectives: first, we aim to formalize the idea of an *Earth  System Data Cube* that is tailored to explore a variety of Earth system data streams together and thus largely complements the existing approaches. The proposed mathematical formalism intends to illustrate how one can efficiently operate such data cubes. Second, the paper aims at introducing the *Earth System Data Lab* (ESDL, https://earthsystemdatalab.net). The ESDL is an integrated data and analytical hub that curates a multitude of data streams representing key processes of the different subsystems of the Earth in a common data model and coordinate reference system. This infrastructure enables researchers to apply their *user defined functions* (UDFs) to these *analysis-ready data* (ARD). Together, these elements minimize the hurdle to co-explore a multitude of Earth system data streams. Most known initiatives intend to preserve the resolutions of the underlying data and facilitate their direct exploitation, like the Earth Server (Baumann et al., 2016) or the Google Earth Engine (Gorelick et al., 2017). The ESDL, instead, is built around singular data cubes on common spatiotemporal grids, that include a high number of variables as a dimension in its own right. This design principle is thought to be advantageous compared to building data cubes from individual data streams without considering their interactions from the very beginning.  Due to its multivariate structure and the easy-to-use interface, the ESDL is well-suited for being part of data-driven challenges, as regularly organized by the machine learning community, for example.

The reminder of the paper is organized as follows: Sect. 2 introduces the concept based on a formal definition of *Earth System Data Cubes* and explains how user defined functions can interact with them. In Sect. 3, we describe the implementation of the *Earth System Data Lab* in the programming language Julia and as a cloud based data hub. Sect. 4 then illustrates three research use cases that highlight different ways to make use of the ESDL. We present an example from an univariate analysis, characterizing seasonal dynamics of some selected variables; an example from high-dimensional data analysis; and an example

for the representation of a model-data-integration approach. In Sect. 5, we discuss the current advantages and limitations of our approach and put an emphasis on required future developments.

**2 Concept**

85 Our vision is that multiple spatiotemporal data streams shall be treated as a singular, yet potentially very high-dimensional data stream. We call this singular data stream an *Earth System Data Cube*. For the sake of clarity, we introduce a mathematical representation of the Earth  System Data Cube and define operations on it. Further details on an efficient implementation are provided in Sections 3.2 and 3.3.

Suppose we observe $p$ variables $Y^1, \ldots, Y^p$, each under a (possibly different) range of conditions. A first step towards data

90 integration is to (re)sample all data streams onto a common domain $J$ (e.g., a spatiotemporal grid) to obtain the indexed set $\{(Y_j^1, \ldots, Y_j^p)\}_{j \in J}$ of multivariate observations. Observations obtained from different variables are then identified as different coordinates in the multivariate array $Y$. However, when calculating simple variable summaries, or performing  spatiotemporal aggregations of the data,  such a representation can be computationally obstructive. We therefore propose to consider the "vari-

95 able indicator" $k \in \{1, \ldots, p\}$ as simply another dimension of the index set, and view the data as the collection $\{X_i\}_{i \in I}$ of *univariate* observations, where $I = J \times \{1, \ldots, p\}$  and where $X_{(j,k)} := Y_j^k$. With this idea in mind, we now formally define the *Earth System Data Cube* (short *data cube*).

A data cube $C$ consists of a triplet $(L, G, X)$ of components to be described below.

– $L$ is a set of labels, called dimensions, describing the axes of the data cube. For example, $L = \{lat, lon, time, var\}$ de-
100 scribes a data cube containing spatiotemporal observations from a range of different variables. The number of dimensions $|L|$ is referred to as the *order* of the cube $C$, in the above example, $|L| = 4$.

– $G$ is a collection $\{\mathrm{grid}(\ell)\}_{\ell \in L}$ of grids along the axes in $L$. For every $\ell \in L$, the set $\mathrm{grid}(\ell)$ is a discrete subset of the domain of the axis $\ell$, specifying the resolution at which data is available along this axis. Every set $\mathrm{grid}(\ell)$ is required to contain at least two elements. Dimensions containing only one grid point are dropped. The collection $G$ defines the
105 hyperrectangular index set $I(G) := \bigtimes_{\ell \in L} \mathrm{grid}(\ell)$ , motivating the name "cube". For example,

$$
\begin{aligned}
I(G) &= \bigtimes_{\ell \in L} \mathrm{grid}(\ell) \\
&= \mathrm{grid}(lat) \times \mathrm{grid}(lon) \times \mathrm{grid}(time) \times \mathrm{grid}(var) \\
&= \{-89.75, \ldots, 89.75\} \times \{-179.75, \ldots, 179.75\} \times \{01.01.2010, \ldots, 31.12.2010\} \times \{\mathrm{GPP}, \mathrm{SWC}, \mathrm{R}_g\} \\
&= \{(-89.75, -179.75, 01.01.2010, \mathrm{GPP}), \ldots, (89.75, 179.75, 31.12.2010, \mathrm{R}_g)\}.
\end{aligned}
$$

110 Since $G$ and $I(G)$ are in one-to-one correspondence, we will use the two interchangeably.
* * *
[1]The symbol $\bigtimes$ indicates a cartesian product.

- $X$ is a collection of data $\{X_i\}_{i \in I(G)} \subseteq \mathbb{R}_{\mathrm{NA}} := \mathbb{R} \cup \{\mathrm{NA}\}$ observed at the grid points in $I(G)$. Here, "NA" refers to "not available".

In this view, the data can be treated as a collection $\{X_i\}_{i \in I(G)}$ of *univariate* observations, even if they encode different variables. In the above example the variable axis is a nominal grid with the entries GPP (gross primary production), SWC (soil water content), and $R_g$ (global radiation). The set of all data cubes with dimensions $L$ will be denoted by $\mathcal{C}(L)$. Data cubes that contain one variable only  can be considered as special case; other common choices of $L$ are described in Table 1. The list of example axes labels used in the table is, of course, not exhaustive. Other relevant dimensions could be, for example, model versions, model parameters, quality flags, or uncertainty estimates. Note that *by definition*, a data cube only depends on its dimensions through the *set* of axes $L$, and is therefore indifferent to any order of these. In the remainder of this article, the notion of data cubes refers to this concept. Note that dropping dimensions that only contain one grid point is not the only possible way of working with data cubes. Another equally valid idea is to maintain grids of length one and integrate them to the workflow.

[revised manuscript text omitted]

| | | | | |
|---|---|---|---|---|
| • Selected data sets
• Sourcing data sets | • Data readers
• Pre-processing
• Common grids
• Chunking
• Metadata | • Cubes in the cloud
• Computing resources
• Distribution | • Data access and
analysis API for
Python and Julia
(and R) | • Users implementing
diverse use cases
• Sharing examples |

**Figure 2.**  Workflow putting the  ESDL concept into practice: selected data  sets are preprocessed to  common grids and  saved in cloud ready data formats (zarr).  Based on these cubed data sets,  a global Earth System Data Cube can be produced that is  either stored locally or in  the  cloud. Via appropriate application programming interfaces (APIs) users can efficiently access the  ESDC in their native language. Users can fully focus on designing user defined functions and workflows.

**3 Data streams and implementation**

The concept as described in Sect. 2 is generic, i.e. independent of the implemented Earth  System Data Cube and of the technical solution of the implementation. Fig. 2 shows how the concept outlined above is realized from a practical point of view. The flowchart shows that the starting point is the collection of relevant data streams which then need to be preprocessed in order to be interpretable as a single data cube. The ESDC itself may be stored locally or in the cloud and can be accessed from various users simultaneously based on different APIs. In the following, we firstly present the data used in our implementation of the ESDL which is available online, and secondly describe the implementation strategy for the API we developed in this project.

**3.1 Data streams in the ESDL**

The data streams included so far were chosen to enable research on the following topics (a complete list is provided in Appendix A):

(i) *Ecosystem states* at the global scale in terms of relevant biophysical variables. Examples are, for instance, leaf area index (LAI), the fraction of photosynthetically active radiation (FAPAR), and albedo (Disney et al., 2016; Pinty et al., 2006; Blessing and Löw, 2017).

(ii) *Biosphere-atmosphere interactions* as encoded in land fluxes of $CO_2$ i.e. GPP, terrestrial ecosystem respiration ($R_{eco}$), and the net ecosystem exchange (NEE) as well as the latent heat (LE) and sensible heat (H) energy fluxes. Here, we rely mostly on the FLUXCOM data suite (Tramontana et al., 2016; Jung et al., 2019).

(iii) *Terrestrial hydrology* requires a wide range of variables. We mainly ingest data from the Global Land Evaporation Amsterdam Model (GLEAM; Martens et al., 2017; Miralles et al., 2011) which provides a series of relevant surface hydrological properties such as surface (SM) and root-zone soil moisture (SMroot), but also potential evaporation (Ep) and evaporative stress (S) conditions, among others. Ingesting entire products such as GLEAM ensures internal consistency.

(iv) *State of the atmosphere* is described using data generated by the Climate Change Initiative by the ESA (CCI) in terms of aerosol optical thickness at different wavelength (AOD550, AOD555, AOD659, and AOD1610; Holzer-Popp et al., 2013), total ozone column (Van Roozendael et al., 2012; Lerot et al., 2014), as well as surface ozone (which is more relevant to plants), and total column water vapour (TCWV; Schröder et al., 2012; Schneider et al., 2013).

(v) *Meteorological conditions* are described via the reanalysis data i.e. the ERA5 product. Additionally, precipitation is ingested from the Global Precipitation Climatology Project (GPCP; Adler et al., 2003; Huffman et al., 2009).

Together, these data streams form data cubes of intermediate spatial and temporal resolutions (0.25°, 0.083°; both 8-daily), visualized in Fig. 3. These variables described here are described in more detail in a list provided in Appendix A, which may, however, already be incomplete at the time of publication, as the ESDL is a living data suite, constantly expanding according to users' requests. For the latest overview, we refer the reader to the website (https://www.earthsystemdatalab.net/). Note that we have not considered the integration of uncertainty as another dimension in the current implementation. The rationale is that each of the data products comes with a specific uncertainty flag or estimate that cannot be merged in an own dimension. This is an open aspect that needs to be addressed in future developments.

Moreover, to To show the portability of the approach, we have developed a regional data cube for Colombia. This work supports the Colombian Biodiversity Observational Network activities within GEOBON. This regional data cube has a 1km (0.083°) resolution and focuses on remote sensing derived data products (i.e. LAI, FAPAR, the normalized difference vegetation index NDVI, the enhanced vegetation index EVI, LST, and burnt area). In addition to the global ESDL, monthly mean products such as cloud cover (Wilson and Jetz, 2016) have been ingested given of their recurrent applicability in biodiversity studies at regional scales. Data layers from governmental organizations providing detailed information about ecosystems

[Figure]

**Figure 3.** Visualization of the implemented Earth System Data Cube (an animation is provided online https://youtu.be/9L4-fq48Ev0). The figure shows from the top left to bottom right the variables sensible heat (H), latent heat (LE), gross primary production (GPP), surface moisture (SM), land surface temperature (LST), air temperature (Tair), cloudiness (C), precipitation (P), and water vapour (V). References to the individual data sources are given in Appendix 1. Here, the resolution in space is 0.25°, in time 8-days, and we are inspecting the time from May 2008 to May 2010, the spatial range is from 15°S to 60°N, and 10°E to 65°W.

are also available that allow a national characterization and deeper understanding of ecosystem changes by natural or human drivers. These are the national ecosystem map (IDEAM et al., 2017), biotic units map (Londoño et al., 2017), wetlands (Flórez et al., 2016) and agriculture frontier maps (MADR-UPRA, 2017). Additionally, GPP, evapotranspiration, shortwave radiation, PAR and diffuse PAR from the Breathing Earth System Simulator (BESS; Ryu et al., 2011; Jiang and Ryu, 2016b; Ryu et al., 2018) and albedo from QA4ECV (http://www.qa4ecv.eu/) are available, among others. This regional Earth  System Data Cube should serve as a platform for analysis in a region with variability of landscape, high biodiversity, ecosystem transitions gradients and facing rapid land use change (Sierra et al., 2017).

**3.2 Implementation**

To put the concept of an Earth  System Data Cube as outlined in Sect. 2 into practice,  we need suitable access APIs (see Fig. 2). A co-author of this paper (FG) developed  one API in the relatively young scientific programming  language Julia (julialang.org; Bezanson et al., 2017)  which is provided via the `ESDL.jl` package.  Additionally, all functionalities are also available in Python based on existing libraries and documented online.

250 In both cases, the goal was that the user does not have to explicitly deal with the complexities of sequential data input/output handling and can concentrate on implementing the atomic functions and workflows, while the system takes care of necessary out of core and out of memory computations. The following is a sketched description of the principles of the Julia-based `ESDL.jl` implementation. We choose Julia to translate the concepts outlined into efficient computer code because it has clear advantages for data cube applications besides its general elegance in scientific computing in terms of speed, dynamic program-

255 ming, multiple dispatch, and syntax (Perkel, 2019). Specifically, Julia allows for generic processing of high-dimensional data without large code repetitions. At the core of the Julia `ESDL.jl` toolbox are the `mapslices` and `mapCube` functions, which execute user-defined functions on the data cube as follows:

- Given some large data cube $C = (L, G, X)$, the `ESDL` function `subsetcube(C)` will retrieve a handle to $C$ that fully describes $L$ and $G$.

260 - Knowledge on the desired $L$ and $G$ allows us to develop a suitable user defined function $f_E^R$.

- Depending on the exact needs, `mapslices` and `mapCube` will then be used to apply the $f_E^R$ on a cube as illustrated in Fig. 1. `mapCube` is a strict implementation of the cube mapping concept described here, where it is mandatory to explicitly describe $E$ and $R$ such that the atomic function is fully operational. `mapslices` is a convenient wrapper around the `mapCube` function that tries to impute the output dimensions given the user function definition to ease the

265 application of the functions where the output dimensions are trivial. Internally, `mapslices` and `mapCube` verify that $E \subseteq L$ and other conditions.

The case studies developed in Sect. 4 are accompanied with code that illustrates this approach in practice.

Of course there are also alternatives to Julia. Lu et al. (2018) recently reviewed different ways of applying functions on array data sets in R, Octave, Python, Rasdaman and SciDB. One requirement of such a mapping function is that it should be scalable,

270 which means that it should process data larger than the computer memory and, if needed, in parallel. While existing solutions are sufficient for certain applications, most are not consistent with the cube mapping concept as described in Sect. 2. For instance, the required handling of complex workflows of multiple cubes (Eq. 8) is typically not possible in the existing solutions that have been reviewed. In some cases, issues in the computational efficiency of the underlying programming languages render certain solutions not suitable. This is particular the case when user-defined functions become complex. Likewise, certain

275 properties such as the desired indifference to the ordering in axes dimensions are often not foreseen. One suitable alternative to Julia is available in Python. The `xarray` (http://xarray.pydata.org) and `dask` packages have been successfully utilised in the Open Data Cube, Pangeo, and xcube initiatives. An Extensive descriptions on how to work in the ESDL with both Python and Julia can be accessed from the website: earthsystemdatalab.net.

The open source implementation of the ESDL also implies that one can easily extend the stored data sets. The online

280 documentation shows in detail how additional data can be added to the ESDL. In particular, if the data share common axes and are stored in a compatible format (as described below in Sect. 3.3) this does not require major efforts.

**3.3 Storage and processing of the data cube**

[revised manuscript text omitted]

415   The question on the underlying dimensionality could also be interrogated in a different way. While this study investigates the intrinsic dimensionality locally, i.e., along the dimensions latitude and longitude, another recent study based on the ESDL by Kraemer et al. (2019) used a global PCA. Each observation is a point with coordinates $lat, long, time,$ and the aim is to compress the $var$ dimension. The form of the analysis is the following,

$$f_{\{var\}}^{\{princomp\}} : \mathcal{C}(\{lat,lon,time,var\}) \to \mathcal{C}(\{lat,lon,time,princomp\}), \tag{14}$$

420   and was applied to a subset of ESDL variables that describe dynamics in terrestrial ecosystems. This study corroborates the idea that land surface dynamics can be well represented in a surprisingly low-dimensional space. The analysis presented by

[Figure]

**Figure 7.** Global patterns of locally estimated temperature sensitivities of ecosystem respiration $Q_{10}$, a) via a conventional parameter estimation approach, and b) via a time-scale dependent parameter estimation method. The latter reduce the confounding influence of seasonality and lead to a fairly homogeneous map of temperature sensitivity.

Kraemer et al. (2019) suggests globally a much lower intrinsic dimensionality of three compared to what we find here based on a grid-cell level analysis. This number corresponds to areas that are marked by a strong seasonality in our case. This is plausible, because the areas that show high intrinsic dimensionality in Fig.5 are those where seasonal variability is low compared to the high-frequency variability (Linscheid et al., 2019). Local effects of this kind vanish when all spatial points are jointly analyzed.

**4.3 Model-parameter estimation in the ESDL**

Another key element in supporting Earth system sciences with the ESDL (and related initiatives) is to enable model development, parametrization, and evaluation. To explore this potential we present a parameter estimation study that  considers two variables only, but it helps to illustrate the approach. In fact, the approach could be extended to exploit multiple data streams in complex models. The example presented here quantifies the sensitivities of ecosystem  natural release of $CO_2$ by  ecosystems—to fluctuations in temperature. Estimating such sensitivities is key for understanding and modelling the global climate-carbon cycle feedbacks (Kirschbaum, 1995). The following simple model (Davidson and Janssens, 2006) is widely used as a diagnostic description of this process:

$$R_{\text{eco},i} = R_{\text{b}} Q_{10}^{\frac{T_i - T_{\text{ref}}}{10}}, \tag{15}$$

where $R_{\text{eco},i}$ is ecosystem respiration at time point $i$ and the parameter $Q_{10}$ is the temperature sensitivity of this process, i.e. the factor by which $R_{\text{eco},i}$ would change by increasing (or decreasing) the temperature $T_i$ by $10°$. An indication of how much respiration we would expect at some given reference temperature $T_{\text{ref}}$ is given by the pre-exponential factor $R_{\text{b}}$. Under this model, one can directly estimate the temperature sensitivities from some observed respiration and temperature time series.

440  Technically this is possible and Eq. (16) describes a parameter estimation process as an atomic function,

$$f_{\{time,var\}}\underline{^{\{para\},\{time\}}}\underwave{^{\{par\},\{time\}}} : \mathcal{C}(\{lat,lon,time,var\}) \rightarrow \mathcal{C}(\{lat,lon,par\}) \underwave{\times \mathcal{C}(\{lat,lon,time\})}, \quad (16)$$

that expects a multivariate time series, and returns a parameter vector. Figure 7a visualizes these estimates, which are comparable to many other examples in the literature (see e.g. Hashimoto et al., 2015) and depict pronounced spatial gradients.  High latitude ecosystems seem to be particularly sensitive to temperature variability according to such an analy-

445  sis.

However, it has been shown theoretically (Davidson and Janssens, 2006), experimentally (Sampson et al., 2007), and using model-data fusion (Migliavacca et al., 2015), that the underlying assumption of a constant base rate is not justified. The reason is that the amount of respirable carbon in the ecosystem will certainly vary with the supply, and hence phenology, as well as with respiration limiting factors such as water stress (Reichstein and Beer, 2008). In other words, ignoring the seasonal time

450  evolution of $R_{\mathrm{b}}$ leads to substantially confounded parameter estimates for $Q_{10}$.

One generic solution to the problem is to exploit the variability of respiratory processes at short-term modes of variability. Specifically, one can apply a timescale dependent parameter estimation (SCAPE; Mahecha et al., 2010b), assuming that $R_{\mathrm{b}}$ varies slowly e.g. on a seasonal and slower timescale. This approach requires some time series decomposition as described in Sec. 4.2. The SCAPE idea requires to rewrite the model, after linearization, such that it allows for a time-varying base rate,

455  $$\ln R_{\mathrm{eco},i} = \ln R_{\mathrm{b},i} + \frac{T_i - T_{\mathrm{ref}}}{10} \ln Q_{10}. \quad (17)$$

The discrete spectral decomposition into frequency bands of the log-transformed respiration allows to estimate $\ln Q_{10}$ on specific timescales that are independent of phenological state changes  Mahecha et al., 2010b Conceptually, the model estimation process now involves two steps (Eqs. (18) and (19)), a spectral decomposition where we produce a data cube of higher order,

460  $$f_{\{time\}}^{\{time,freq\}} : \mathcal{C}(\{lat,lon,time,var\}) \rightarrow \mathcal{C}(\{lat,lon,time,var,freq\}) \quad (18)$$

followed by the parameter estimation, which differs from the approach described in Eq.16, as this approach only returns a singular parameter ($Q_{10}$), whereas $\ln R_{\mathrm{b},i}$ now becomes a time series:

$$f_{\{time,var,freq\},}^{\{\},\{time\}} : \mathcal{C}(\{lat,lon,time,var,freq\}) \rightarrow \mathcal{C}(\{lat,lon\}) \times \mathcal{C}(\{lat,lon,time\}) \quad (19)$$

The results of the analysis are shown in Fig. 7b where we find generally a much more homogeneous and better constrained

465  spatial pattern of $Q_{10}$. As suggested in the site-level analysis by Mahecha et al. (2010b) and later by others (see e.g. Wang et al., 2018) we find a global convergence of the temperature sensitivities. We also find that e.g. semi-arid and savanna-dominated regions clearly show lower apparent $Q_{10}$ (Fig. 7a) compared to the SCAPE approach (Fig. 7b). Discussing these patterns in detail is beyond the scope of this paper, but in general terms these finding are consistent with the expectation that in  semi-arid ecosystems confounding factors act in the opposing direction (Reichstein and Beer, 2008).

470  From a more methodological point of view this research application shows that it is well possible to implement a multistep analytic workflow in the ESDL that combines time series analysis and parameter estimation. Once the analysis is implemented,

it requires essentially  two sequential atomic functions. The results obtained have the form of a data cube and could be integrated into subsequent analyses. Examples include comparisons with in-situ data, eco-physiological parameter interpretations or assessment of parameter uncertainty in more detail. As mentioned above, this case study only considers two variables and thereby does not exploit the wider multivariate potential of the ESDL. The example of temperature sensitivity could easily be combined with further estimations of water stress, linked to primary production, or even become part of a simple terrestrial surface scheme.

**4.4 Bivariate relations in vector-cubes**

The original idea of the data cube concept emerged from the need of working with large multivariate gridded data sets. However, the idea of data cubes can be possibly extended to other types of geographical data. One example are vector data cubes, where e.g. polygons form an axis in their own right and each polygon points to a complex spatial shape. Consider, for instance, the need for statistical inferences on the spatial polygons often used in IPCC reports. One relevant question is, for example, understanding the relations of GPP and surface moisture. Fig 8 shows the bivariate histograms between both variables within a selected set of regions. This analysis clearly shows that in many regions of the world, GPP and surface moisture are strongly coupled. Examples are e.g. Central America/Mexico (CAM), North-Earth Brazil (NEB), West Africa (WAF), Southern Africa (SAF), East Africa (EAF), South Asia (SAS), or South Australia/New Zealand (SAU). All of these regions contain significant fractions of semi-arid climates, which can explain the constraints that water availability has on photosynthetic $CO_2$-uptake. In other regions, this relation is less obvious and often not pronounced, probably because the cases of water shortage are rare compared to the normal dynamics that might be constrained by other factors such as temperature. From a computational point of view, this example follows a very different logic, compared to the concept of applying an UDF on some of the cube axis. Rather, this example was computed using an "online" approach which sequentially updates some statistics (here the bivariate histograms) over a given class (here IPCC regions). Such an approach allows calculations with large amounts of data and shows that the ESDL framework can also be coupled with conceptually very different analytical frameworks that might be particularly relevant when working with living data, i.e. with data streams that are constantly updated. In these cases, it is not desirable to constantly re-estimate all relevant quantities across the entire data cube.

**5 Discussion**

In the following, we describe the insights gained during the development of the concept and the implementation of the ESDL, addressing issues arising and critiques expressed during our community consultation processes. We also briefly discuss the ESDL in the light of other developments in the field. Finally, we highlight some challenges ahead and proposed future applications.

[Figure]

**Figure 8.**  Bivariate histograms summarizing the joint distribution of  surface moisture and gross primary production. The  estimates are computed over the  entire time series for the different IPCC regions. The density is square root transformed emphasize areas of higher density. In arid regions (e.g. CAM, NEB, WAF, SAFM, EAF) the tight relation between surface water and primary production is evident.

**5.1 Insights and critical perspectives**

During a community consultation process across various workshops and summer schools, users expressed confusion about the equitable treatment of data cube dimensions (Sect. 2). Considering that an unordered nominal dimension of "variables" is a

dimension as "time" or "latitude" seems counterintuitive at first glance.  Also, concerns have been expressed about whether "time" can be treated analogously to e.g. "latitude". Our main argument during the development of the ESDL was that it is possible, as long as the UDFs are not applied to dimensions where they would produce nonsense results. But the practical arguments for a common interface prevail. Also, and this is key, the concept and implementation are sufficiently flexible to allow users to  deploy more classical approach to deal with  such data, e.g., analyzing variables separatly, or writing specific UDFs that specifically require spatial or temporal dimensions. However, for research examples structured like the second use case (Sect. 4.2),  the proposed approach was key as it is allowed to efficiently navigate through the variable dimension. It is obviously irrelevant to algorithms of dimensionality reduction, which dimension is compressed and we could have equally asked the question in time domain or across a spatial dimensions, which relates to the well-known empirical orthogonal functions (EOFs) as used in climate sciences (Storch and Zwiers, 1999). In exploratory approaches of this kind, where there is no prior scientific basis for presupposing where the "information-rich zones" are in the data cube, a dimension-agnostic approach clearly pays off. We also favour this idea as it is in-line with other approaches discussed in the community. For instance, the "Data Cube Manifesto" (Baumann, 2017) states that *"Datacubes shall treat all axes alike, irrespective of an axis having a spatial, temporal, or other semantics."*, a principle that we have radically implemented in the `ESDL.jl` Julia package (Sect. 3). The flexibility we gain is that we are, in principle, prepared for comparable cases where one has to deal with e.g. multiple model versions, model ensemble members, or model runs based on varying initial conditions.

One of the most commonly expressed practical concerns is the choice of  a unique data grid. The curation of multiple data streams within such a data cube grid requires that many data have to undergo reformatting and/or remapping. Of course, this can be problematic at times, in particular when data have been produced for a given spatial or temporal resolution and cannot be remapped without violating basic assumptions. For instance keeping mass balances, integrals of flux densities, and global moments of intensive properties as consistent as possible should always be a priority. However, for the data cube approach implemented here we decided to accept certain simplifications. The availability of a multitude of relevant data to study Earth system dynamics is a key incentive to use the ESDL and goes much beyond many disciplinary domains. But, as we have learned in this discussion, it comes at the price of some pragmatic trade-offs. A fundamental advancement of our approach would be to natively deal with data streams from unequal grids.

 The current notation of the concept has been criticized for being unsuitable for dealing with so-called vector data cubes (Pebesma and Appel, 2019). Indeed other conceptual approaches are more suited than ours to treat such examples (see e.g. Gebbert et al., 2019). But the research example briefly described in Sect. 4.4 and Fig. 8 does showcases such a possibility. In this case, the idea of mapping a single function across some dimensions cannot be trivially realized, but it opens novel perspectives to compute statistics based on very big data. Further research needs to be done on developing the ESLD in such directions because it would allow not only for dealing with big data issues, but also to update statistics without having to recompute data processed in earlier steps. This can solve the challenges of dealing with "living data".

One of the main concerns expressed by users, in particular by 30 young researchers who participated in the project during an Early Adopter phase, is the demand for latest data in the ESDL. This is why the concept presented here and its implementation

540 should be further developed into a persistent infrastructure. Such a step is challenging and there is a trade-off  to be made between wishing to include latest data streams (ideally even in near real time),  versus constantly expanding the access API and portfolio of example workflows. The ESDL thus depends on the enduring enthusiasm of the user community and funding agencies to support the idea in this respect and grow steadily into new domains, help us adding data streams, and  actively co-develop the approach.

545 **5.2   Relation to other initiatives and platforms**

Over the past few years, several initiatives, platforms, and software solutions (Lu et al., 2018; Sudmanns et al., 2019) have emerged based on similar considerations as those motivating the Earth System Data Lab. Some of these platforms and software solutions are explicitly constructed around the idea of data cubes (e.g. Baumann et al., 2016; Lewis et al., 2017; Appel and Pebesma, 2019). Nevertheless, the concept of "data cube" is still not fully consolidated in the Earth system science.

550  It was only in 2019  the Open Geospatial Consortium (OGC)  opened a public discussion towards establishing standards for data cubes.

[revised manuscript text omitted]
 fact, the ESDL approach is perfectly suited to handle e.g. the output of the actual CMIP data as we have already exemplified[2]. Of course, any other model ensembles can be treated analogously.

In terms of application domains we see high potential in the following areas:

- *Human-environment interactions:* Addressing the complexities of "human-environmental interactions" (Schimel et al., 2015) is a particular challenge. Making the ESDL fit for this purpose would require integrating a variety of (at least) spatially explicit population estimates (Doxsey-Whitfield et al., 2015) and socioeconomic data Smits and Permanyer (2019). The latter represent a fundamentally novel development that has great potential for understanding e.g. dynamics of disasters impacts (Guha-Sapir and Checchi, 2018), among other issues. In fact this integration is a grand challenge ahead (Mahecha et al., 2019), but not out of reach for the ESDL.

- *Biodiversity research:* Another question of high societal relevance is to understand how patterns of biodiversity affect ecosystem functioning (Emmett Duffy et al., 2017; García-Palacios et al., 2018). In the light of a global decline in species richness (cf. latest global reports https://www.ipbes.net/), this question is of uttermost importance. The ESDL is only partly fit for this purpose, as it would require the ingestion of a wide range of essential biodiversity variables (Pereira et al., 2013; Skidmore et al., 2015), beyond the ones we have already available. But still, the ESDL is conceptually prepared to deal with these challenges (compare e.g. the demands described in Hardisty et al., 2019) and would be particularly suitable for relating biodiversity patters to the so-called ecosystem function properties (Reichstein et al., 2014; Musavi et al., 2015). In fact, in the regional application of the ESDL we have focused on Colombia and its wider region to explore linkages of this kind relying on remote sensing derived variables that are relevant for this context.
* * *
[2]https://gist.github.com/meggart/2d544be2c1368f8774d0a21ea4633985

635    – *Oceanic sciences:* Extending the ESDL for ocean data is desired and conceptually possible. Surface parameters, e.g.  phytoplankton phenology derived from remote sensing (Racault et al., 2012), can be treated analogously to terrestrial surface parameters. Other dynamics, e.g. the analysis and exploration of ocean-land coupling mechanisms, ocean-atmosphere interactions, and land-atmosphere interactions triggered by ocean circulation dynamics could in principle be facilitated via the ESDL but require to either vertical or lateral dynamics.

640    – *Solid Earth:* The step towards global, fully data informed, model data is also made in geophysics. For instance, recently Afonso et al. (2019) used an inversion approach to develop a 3D model that fully describes multiple parameters in the Earth interior, including e.g. crustal and lithospheric thickness, average crustal density, and a depth-dependent density of the lithospheric mantle, among other variables. They proposed a tool allowing for inspecting the data interactively at a spatial resolution of $2° \times 2°$ grid in different depth. Clearly, in this case other dimensions are relevant

645    , but the principle  remains the same and, in fact, can be treated in a very similar manner. Future model-data assimilation approaches of this kind could be performed in the context of the ESDL, as well as the aforementioned machine learning for the solid Earth (Bergen et al., 2019).

In summary, we have demonstrated that the ESDL is a flexible and generic framework that can allow various different communities to explore and analyse large amounts of gridded data efficiently. Thinking about the potential paths ahead, the
650 ESDL could become a valuable tool in various fields of Earth system sciences, biodiversity research, computer sciences and other branches of science. The widespread social and political uptake of the concept of planetary boundaries (Rockström et al., 2009; Steffen et al., 2015) underlines the global demand for better quantified process understanding of environmental risks and resource bottlenecks based on empirical evidence. Along these lines, the ESDL concept could be used to address some of the most pressing global challenges. For example, it could become an interface for direct interaction with ECVs, global
655 climate projections and EBVs. Such an interactive interface would allow a much broader community to better understand the data underlying the global assessment reports of the IPCC (Pachauri et al., 2014) and IPBES (Diaz et al., 2019). If coupled to some visual interfaces, the ESDL could also be used by a broader community, enhancing education, communication and decision making process, contributing to knowledge democratization about a deeper understanding of the complex and dynamic interactions in the Earth system.

660 **6   Conclusions**

Exploiting the synergistic potential of multiple data streams in the Earth sciences beyond disciplinary boundaries requires a common framework to treat multiple data dimensions, such as for instance spatial, temporal, variable, frequency and other grids, alike. This idea leads to a data cube concept that opens novel avenues to efficiently deal with data in the Earth system sciences. In this paper, we have formalized the concept of data cubes and described a way to operate on them. The out-
665 lined dimension-agnostic approach is implemented in the Earth System Data Lab that enables users applying a wide range of functions to all thinkable combinations of dimension. We believe that this idea can dramatically reduce the barrier to exploit

Earth system data and serves multiple research purposes. The ESDL complements a range of emerging initiatives that differ in architectures and specific purposes. However, the ESDL is probably the most radical data cubing approach, offering novel opportunities for cross-community data-intensive exploration of contemporary global environmental changes. Future developments in related branches of science and latest methodological developments need to be considered and addressed soon.  At its actual state of implementation, the ESDL  can already contribute to the deeper understanding and more effective implementation of policy-relevant concepts such as the planetary boundaries, essential variables in different subsystems of the Earth, and global assessment reports. We see a particularly high future potential  for data cube concepts as presented for, firstly, interpreting large-scale model ensembles, and secondly, analyzing new multispectral satellite remote sensing data with their constantly increasing spatial, temporal, and spectral resolutions.

*Author contributions.* Conceptual development: M.D.M., F.G. and M.R.; Implementation of the `ESDL.jl` package in the Julia language: F.G.; Implementation of the ESDL: N.F., F.G., M.D.M., G.B.; Notation: R.C., J.P., M.D.M., F.G., G.K., P.P.; Data: D.G.M., M.J.; Paper writing: M.D.M. wrote the manuscript with substantial input from F.G., R.C., J.P. and comments from all co-authors.

*Competing interests.* The authors declare no competing interests.

*Code availability.* All code nessecary to build and analyze the ESDL is available from https://github.com/esa-esdl. The case studies presented in Sect. can be fully reproduced from https://github.com/esa-esdl/ESDLPaperCode.jl.

*Data availability.* All data are available via earthsystemdatalab.net or from the original data providers as indicated in the manuscript.

*Competing interests.* The authors declare no competing interests.

*Acknowledgements.* This paper was funded by the European Space Agency (ESA) via the "Earth System Data Lab" (ESDL) project. All authors thank the ESA and the Integrated Land Ecosystem Atmosphere Processes Study (iLEAPS), Global Research Project for constant support. The implementation of the regional Earth data cube for Colombia was done under the project "Champion user phase; Supporting the Colombia BON in GEO BON" with the ESDL project. The original idea emerged at the iLEAPS–ESA–MPG funded workshop in Frascati 2011 (Mahecha et al., 2011). We thank everyone participating in the various workshops, summer schools, and early adopters, providing

invaluable feedback on the ESDL. We thank everyone who made data freely available such that they could be used in this project. Special thanks to Eleanor Blyth, Carsten Brockmann, Garry Hayman, Toby R. Marthews, and Uli Weber for constant support and critical feedback. We thank Marius Appel, Edzer Pebesma, Alexander Winkler, and two anonymous referees for very helpful comments on the manuscript. 
[revised manuscript text omitted]